**EMBO** *reports*

# Immune cells adapt to confined environments in vivo to optimise nuclear plasticity for migration

Tua Karling & Helen Weavers [ID] [✉]

## Abstract

Cells navigating in complex 3D microenvironments frequently encounter narrow spaces that physically challenge migration. While in vitro studies identified nuclear stiffness as a key rate-limiting factor governing the movement of many cell types through artificial constraints, how cells migrating in vivo respond dynamically to confinement imposed by local tissue architecture, and whether these encounters trigger molecular adaptations, is unclear. Here, we establish an innovative in vivo model for mechanistic analysis of nuclear plasticity as *Drosophila* immune cells transition into increasingly confined microenvironments. Integrating live in vivo imaging with molecular genetic analyses, we demonstrate how rapid molecular adaptation upon environmental confinement (including fine-tuning of the nuclear lamina) primes leukocytes for enhanced nuclear deformation while curbing damage (including rupture and micronucleation), ultimately accelerating movement through complex tissues. We find nuclear dynamics in vivo are further impacted by large organelles (phagosomes) and the plasticity of neighbouring cells, which themselves deform during leukocyte passage. The biomechanics of cell migration in vivo are thus shaped both by factors intrinsic to individual immune cells and the malleability of the surrounding microenvironment.

**Keywords** Confined Cell Migration; Nuclear Lamina Deformation; Micronuclei; In Vivo Live-imaging; Extravasation
**Subject Categories** Cell Adhesion, Polarity & Cytoskeleton; Immunology; Methods & Resources

## Introduction

Cell migration plays key roles in health and disease, underpinning the development, maintenance and repair of most body tissues (Yamada and Sixt, 2019). Many cell types, including immune cells ('leukocytes') and cancer cells, navigate long distances through complex 3D microenvironments. Leukocytes, in particular, move from their hematopoietic origins before circulating in vessels or patrolling in tissues for immune surveillance, and often squeeze across vessel walls in response to injury or infection (Ley et al, 2007). The continuous trafficking of these cells relies on their

remarkable ability to migrate rapidly through narrow spaces within tissues and across cellular (e.g. endothelial) barriers, including those imposed by closely apposed cells or 3D meshworks of extracellular matrix (ECM) (Kameritsch and Renkawitz, 2020; Weigelin et al, 2012; Stoitzner et al, 2002). While some cells can actively forge a migratory pathway through dense tissues by proteolysis, many fast-moving cells (including leukocytes and metastatic cancer cells) rely on non-destructive (amoeboid-like) modes of migration (Lämmermann et al, 2008; Wolf et al, 2013, 2003). Leukocytes must be particularly adept at sampling their microenvironment to choose the path of least resistance and rapidly change their morphology to enable passage through small spaces; such plasticity is likely exploited in numerous inflammatory diseases as leukocytes rapidly infiltrate tight tissue spaces (e.g. within tumours) (Renkawitz et al, 2019).

Our current mechanistic understanding of cell migration under confinement predominantly derives from in vitro studies of cells migrating in fabricated (artificial) 3D environments or under applied mechanical load in vitro (Wolf et al, 2013; Renkawitz et al, 2019; Raab et al, 2016). Seminal in vitro studies have identified the nucleus, classically the largest and stiffest organelle, as playing key rate-limiting roles during confined migration of both immune and cancer cell types (McGregor et al, 2016; Davidson et al, 2014; Wolf et al, 2013). Emerging evidence suggests the nucleus is actively involved in decision-making to select the path of least resistance (Renkawitz et al, 2019). While nuclear deformation in vitro can aid migration in complex environments (Thiam et al, 2016) it may, however, predispose cells to potentially catastrophic events, such as DNA damage and nuclear rupture (Thiam et al, 2016). Cells migrating through confined spaces might rely on protective mechanisms to quickly repair damaged nuclear components (Denais et al, 2016; Raab et al, 2016) or strategies to withstand compressive force, such as the assembly of mechanosensitive cytoskeletal 'cages' (Ju et al, 2024).

Leukocytes (such as neutrophils) must possess an extraordinary flexibility to move rapidly through diverse microenvironments (Kameritsch and Renkawitz, 2020). While in vitro studies are accelerating our understanding of nuclear mechanics during migration, it remains unclear how migrating cells within physiological settings in vivo respond dynamically to confinement imposed by ever-changing local tissue architecture, and whether they initiate specialised molecular adaptations for enhanced nuclear plasticity. While adaptations arise early during neutrophil differentiation to give the characteristic lobulated, malleable nucleus (Rowat et al, 2013; Olins et al, 2008; Zwerger et al, 2008; Shin et al,

School of Biochemistry, Biomedical Sciences, University of Bristol, Bristol BS8 1TD, UK. ✉E-mail: helen.weavers@bristol.ac.uk

2013) including alterations in nuclear architecture or chromatin structure (Liu et al, 2023; Olins et al, 2001; Rowat et al, 2013; Zhang et al, 2015), beyond these early developmental events, our molecular understanding of how cells respond dynamically to local confinement remains limited. Do cells adapt their nuclear mechanics as they move between different microenvironments to optimise migration? What are the long-term consequences of physiological confinement on nuclear health and cell function?

These phenomena have remained largely elusive due to a lack of suitable experimental models for integrating long-term live-imaging of cellular dynamics with cell type-specific mechanistic studies, within in vivo settings where cells rapidly transition between environments imposing different physical constraints. Migrating cells in vivo also face mechanical challenges that cannot be fully recapitulated in vitro, as they are frequently constrained by interactions with nearby cells or tissues that possess their own varying material properties; do the relative physical properties (e.g. stiffness) of interacting cells determine which cells ultimately deform to enable passage? Confined cell migration in vivo thus requires complex inter-tissue communication, which is challenging to accurately model in vitro.

There is a clear need for tractable in vivo models where cell—particularly nuclear—mechanics in confined cell migration can be studied in physiologically-relevant settings. While intravital imaging in mammalian systems is accelerating in vivo analysis of cell migration (Yan et al, 2019; Proebstl et al, 2012; Woodfin et al, 2011), the limited in vivo studies of nuclear deformation to date have predominantly utilised mammalian xenograft or explant models for live-imaging of cell migration in confined cells (e.g. transplanted tumour cells) (Yamauchi et al, 2006; Weigelin et al, 2012; Raab et al, 2016; Thiam et al, 2016; Denais et al, 2016). Nevertheless, invertebrate models (such as *Drosophila* and *C. elegans*) are emerging as alternative experimentally tractable systems to investigate confined cell migrations that occur in vivo during development, including epithelial (border cell) migrations within the developing fly ovary (Penfield and Montell, 2023) or *C. elegans* larval epidermis (Bone et al, 2016), as well as the early dispersal of embryonic macrophages (Belyaeva et al, 2022).

Here, we exploit the unique experimental tractability of *Drosophila* to establish an in vivo model for mechanistic analysis of nuclear plasticity as mature immune cells move dynamically between distinct 3D microenvironments, which pose differing levels of confinement. *Drosophila* are well-established as valuable models to dissect fundamental aspects of immune cell migration, including the role of cytoskeletal remodelling, chemotactic signalling and metabolic adaptations (Weavers et al, 2016a, 2016b; Weavers and Wood, 2016; Yolland et al, 2019; Davis et al, 2015). Crucially, our recent work uncovered that *Drosophila* leukocytes (immune cells or 'hemocytes') migrate within narrow vessel-like channels (within the pupal wing) in vivo and can be triggered to 'extravasate' out across their walls in response to nearby tissue damage (Thuma et al, 2018).

We now integrate state-of-the-art time-lapse imaging, with detailed molecular and genetic analyses, to dissect how leukocytes adapt while navigating between distinct 3D environments in vivo, with a focus on nuclear plasticity. We show leukocytes in vivo display increased nuclear deformation (and accompanying DNA damage) as they transition from unconstrained 3D wing environments into the narrow, convoluted wing 'vessels', with even more

dramatic nuclear deformation during transmigration across vessel walls. Mechanistically, we find nuclear plasticity is accompanied by localised actomyosin contractility and facilitated by key molecular adaptations (including altered nuclear lamina composition) that occur as leukocytes transition into more confined microenvironments. Such fine-tuning of the leukocyte nuclear lamina (with precisely balanced A- and B-type Lamins) not only aids nuclear deformation but limits potentially catastrophic nuclear instability (including nuclear rupture and micronuclei formation). Collectively these molecular adaptations, which prime the leukocyte for increased nuclear plasticity, enable more effective navigation through confined in vivo microenvironments. Surprisingly, we also show that phagocytic vacuoles can physically challenge the nucleus and promote deformation, even in the absence of external confinement. Within these in vivo settings, leukocyte nuclear dynamics are further influenced by the compliance of neighbouring cells and tissues with which the leukocyte interacts, which themselves can deform to accommodate the passing leukocyte.

The biomechanics of cell migration in vivo is thus shaped by a dynamic interplay between the unique cell biology of the individual leukocyte (e.g. nuclear architecture and phagosomal content) and the physical properties (e.g. malleability) of the surrounding microenvironment. Ultimately, our work demonstrates that leukocytes rapidly adapt their morphology and nuclear architecture, depending on local tissue properties, to optimise motility in restricted microenvironments in vivo.

# Results

## *Drosophila* leukocytes exhibit morphological adaptations as they transition into increasingly confined tissue microenvironments in vivo

Pupal wing maturation dramatically transforms the 3D microenvironment in which *Drosophila* leukocytes (innate immune cells termed 'hemocytes') migrate (Fig. 1A–F and Movie EV1) (Thuma et al, 2018). At 18 h APF (after puparium formation), the wing has an open 'sac-like' structure with two epithelial sheets separated by extracellular space, largely filled with fluid called hemolymph (Fig. 1A,Bi). Later, the wing tissue undergoes morphological rearrangements that transform the wing into a flat, bilayered epithelium with a highly stereotypical pattern of discrete vessel-like channels at 40 h APF (schematic, Fig. 1C; histology, Fig. 1Di) (Fristrom et al, 1993). By 75 h APF, the pupal wing is highly folded and the narrow vessel-like channels follow highly convoluted paths (schematic, Fig. 1E; histology, Fig. 1Fi); these narrow channels become the mature 'veins' of the adult wing (Fig. EV1A). Live-imaging of the secreted GFP-tagged protein Apolipoprotein-GFP (that labels the extracellular space) highlighted the contrasting space available for leukocyte migration within the different wing environments (Fig. 1Bii,Dii,Fii). Time-lapse imaging revealed *Drosophila* leukocytes initially migrated within the open, unconstrained extracellular space of the 18 h APF wing (between upper and lower wing epithelia, Fig. 1Biii–iv and Movie EV1) (Weavers et al, 2016b, 2018; Sander et al, 2013). As wing morphogenesis proceeded, these leukocytes became restricted to the more confined spaces within the lumen of the 40 h APF channels (Fig. 1Diii–iv

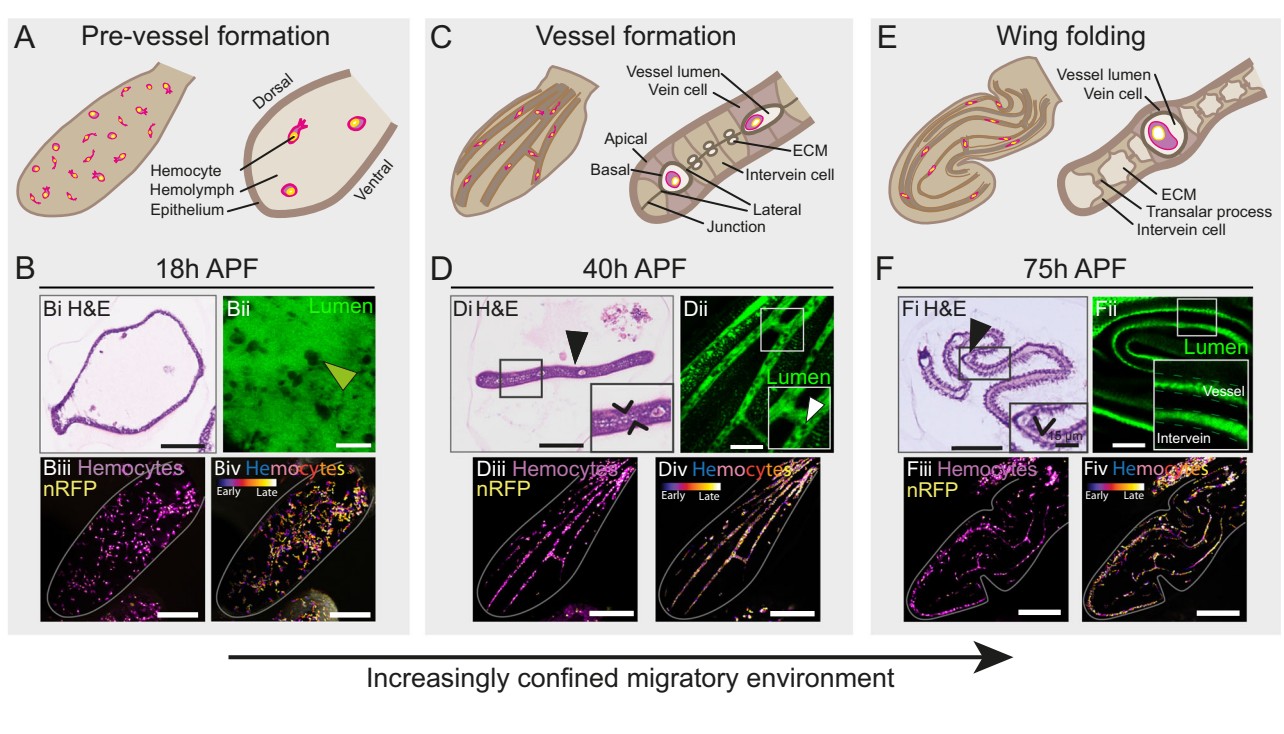

Increasingly confined migratory environment

**A** Pre-vessel formation
**B** 18h APF
**C** Vessel formation
**D** 40h APF
**E** Wing folding
**F** 75h APF

**G** Vessel diameter
**H** Hemocytes at vessel cross-section
**I**
**J** Tracheal network
**K** Nervous system

Live *in vivo* imaging of immune cell morphology

Cell morphology
**L** Hemocytes
**M** **N** **O** **P**

Nuclear morphology
**Q** Hemocyte nuclei
**R** **S**
**T** Hemocyte retention during vessel formation

**Figure 1.  Migrating immune cells in vivo adapt during transition into more confined vessel microenvironments.**

(A–F) Schematics of pupal wing morphology before (A) and after vessel formation (C), and after wing folding (E). Pupal wings at 18 h APF (B), 40 h APF (D) and 75 h APF (F) labelled for Hematoxylin and eosin (i, blue nuclei, pink ECM and cytoplasm), Apolipoprotein-GFP (ii, green), hemocytes (iii, cytoplasm, magenta; nuclei, yellow) and time-projection of hemocyte migration (iv) over 2 h (18 h APF) or 5 h (40 h or 75 h APF); wings outlined in grey. Solid arrowheads indicate hemocytes and open arrowheads indicate intervein space. (G) Vessel diameters (measured using Vkg-GFP, green) at 40 h and 75 h APF (ACV and PCV, anterior and posterior cross veins; LV3-5, lateral cross veins); individual dots are averages of 4 measurements per vein, with large black dots being the average diameter per pupae. (H) Hemocytes (magenta cytoplasm, yellow nucleus), passing each other (arrows indicate migration direction; arrowhead indicates nuclei passing), quantified as numbers of hemocytes across the vessel cross-section in a single timepoint, shown as percentages of all hemocytes across a pupal wing. (I–K) Schematics of wing trachea and nervous system (I); 40 h and 75 h APF pupal wings labelled for trachea (J, cyan) or nerves (K, yellow). Inset of 75 h APF vessel showing hemocyte cell body (Simu-GFP, magenta) deforming (arrowhead) next to a nerve (Elav>Tom, yellow). (L) Hemocyte morphology (grey dashed lines indicate vessel wall) at 18 h, 40 h and 75 h APF. (M–P) Hemocyte cytoplasmic minimum diameter (M), area (N), circularity (O) and average hemocyte migration speed (P) over 20 min. (Q) Example hemocyte nuclei (white) at 18 h, 40 h and 75 APF; minimum nuclear diameter as measured between arrowheads is shown. (R) Average hemocyte nuclear area during a 20 min imaging period. (S) Minimum hemocyte nuclear diameter measured from a single time-point. (T) Hemocyte wing retention from 18 h through 75 h APF; UV-exposed Kaede-expressing hemocytes (insets, hemocytes outlined) captured with 488 nm (unconverted, Kaede) and 561 nm (photoconverted, Kaede*) at 18 h APF and 75 h APF. Genotypes used were w67 (B, D, F), Srp > nRFP; Crq > GFP (B, D, F, L–O, Q, S), Srp > GFP, SrpH2AmChe (H), Btl > GFP (J), Elav>Tom;Simu-GFP (K), Srp > nRFP (P, R) and Srp>Kaede (T). Data information: Scale bars represent 50 µm (Bi, Bii, Di, Dii, Fi, Fii), 200 µm (Biii, Biv, Diii, Div, Fiii, Fiv, J, K), 5 µm (H and Q), 10 µm (K inset and L), 100 µm (T) or 15 µm (T, insets). For (M–P) and (R, S), scatter plots show small dots (hemocytes) and large dots (wing medians), with bar charts showing median and 95% CI. Error bars show mean +/− SD (G, H, T); unpaired t test (G), One-way ANOVA (Kruskal-Wallis) with Dunn's multiple comparisons test (M–P, R, S) or Mann-Whitney U test (T). For (G, H), N = 4 and 5 wings for 40 h and 75 h APF, respectively, with 302 and 147 hemocytes (H). N = 3, 4 and 5 wings (M, S), N = 3, 3 and 5 wings (N, O), N = 5, 4 and 5 wings (P, R), and N = 268, 246 and 127 hemocytes (M), N = 216, 102 and 136 hemocytes (N, O), N = 290, 138 and 98 hemocytes (P), N = 286, 139 and 102 hemocytes (R), and N = 233, 173 and 127 hemocytes (S) for 18 h, 40 h and 75 h APF, respectively. For (T), N = 5 pupae, N = 526 and 268 hemocytes for 18 h and 75 h. Source data are available online for this figure.

and Movie EV1) (Thuma et al, 2018). By 75 h APF, wing leukocytes were found within the lumens of narrower, convoluted wing vessels making the path of immune cell migration even more tortuous (Fig. 1Fiii–iv and Movie EV1); channels within 75 h APF wings were not only more convoluted but narrower than those of 40 h APF wings (Figs. 1G and EV1B), comparable to the smallest post-capillary venules of mammals (Gopalan and Kirk, 2022). While contractile wing hearts establish a modest pulsatile flow of hemolymph through the narrow wing vessels (Tögel et al, 2013) by 75 h APF, we previously demonstrated wing leukocytes can migrate independently of the flow (Thuma et al, 2018).

While vessels of 40 h APF wings were wider than those of 75 h APF (Fig. 1G), leukocytes were more frequently observed crossing paths within 40 h APF vessels (Fig. 1H). By 75 h APF, the vessels were considerably longer and significantly fewer leukocytes were present within vessels than at 18 h or 40 h APF (Fig. EV1C,D), suggesting a similar 'effective space' could be available for individual leukocytes at 40 h and 75 h APF stages. Nevertheless, the leukocyte migratory environment within these channels was further confined by the progressive growth of additional cell types (Figs. 1I and EV1E,F), including oxygen-transporting trachea (Fig. 1J) and neurons (Fig. 1K) (Fristrom et al, 1994; Murray et al, 1984) with which leukocytes were observed to directly interact and navigate around (Fig. 1K, inset or Fig. EV1F).

Live time-lapse imaging of cellular dynamics revealed that leukocytes exhibited altered morphology and behaviour as the wing microenvironment became increasingly restrictive (Fig. 1L–S and Movie EV2). Leukocytes migrating within the less constrained 18 h APF wings were generally large in size (diameter and area, Figs. 1M,N and EV1I) and predominantly round (Figs. 1O and EV1G,H). In contrast, leukocytes migrating under progressively more physiological confinement within the narrow, convoluted channels were significantly reduced in size (Figs. 1M,N and EV1I) and exhibited significantly more elongated morphology (Figs. 1O and EV1G,H). Upon transition to confinement at 40 h APF, leukocytes also moved marginally slower than when migrating in the open 18 h APF environment (Fig. 1P). Leukocyte

cytoplasmic morphological changes were accompanied by striking nuclear changes, including significant reductions in nuclear diameter and area (Figs. 1Q–S and EV1J). Long-term tracking of wing leukocytes using the photoconvertible fluorophore Kaede demonstrated that the same leukocyte population persists within the wing tissue from 18 h to 75 h APF (Figs. 1T and EV1K,L); while photoconverted Kaede could be transmitted to daughter leukocytes following proliferation, minimal leukocyte proliferation was observed as vessels formed within the wing microenvironment (Fig. EV1M). Thus migrating wing leukocytes might adapt their cellular and nuclear morphology in vivo to optimise migration as the wing environment matures and becomes increasingly confined.

## Increasing environmental confinement in vessel-like channels in vivo triggers enhanced leukocyte nuclear dynamics

Our data suggest Drosophila wing leukocytes could offer mechanistic insight into cellular adaptation to migration within increasingly confined microenvironments in vivo. Here, live-imaging of cellular dynamics at high spatio-temporal resolution revealed that leukocyte nuclei were highly dynamic during migration within vessel-like channels in vivo, undergoing frequent reversible cycles of deformation (Fig. 2A,B and Movie EV3). To comprehensively quantify nuclear dynamics, we developed a computational pipeline that robustly identified leukocyte nuclei within 4D imaging data (Figs. 2C, EV2A and Movie EV4) and quantified numerous cellular features (including nuclear Shape Factor, Fig. 2C). Shape Factor (SF) is a measurement of elongation, where a value close to 0 reflects an isotropic, rounded shape and a value close to 1 implies an elongated shape (Fig. EV2B) (Olenik et al, 2023). Nuclei of leukocytes migrating within vessels (at 40 h and 75 h APF) were significantly more elongated (with a higher mean SF, Fig. 2D and maximum SF, Fig. 2E) than the nuclei of leukocytes migrating within the less constrained 18 h APF wings. Nuclei of leukocytes migrating within vessels were more dynamic, with more frequent deformations (Fig. 2F) and consequently, they exhibited a greater

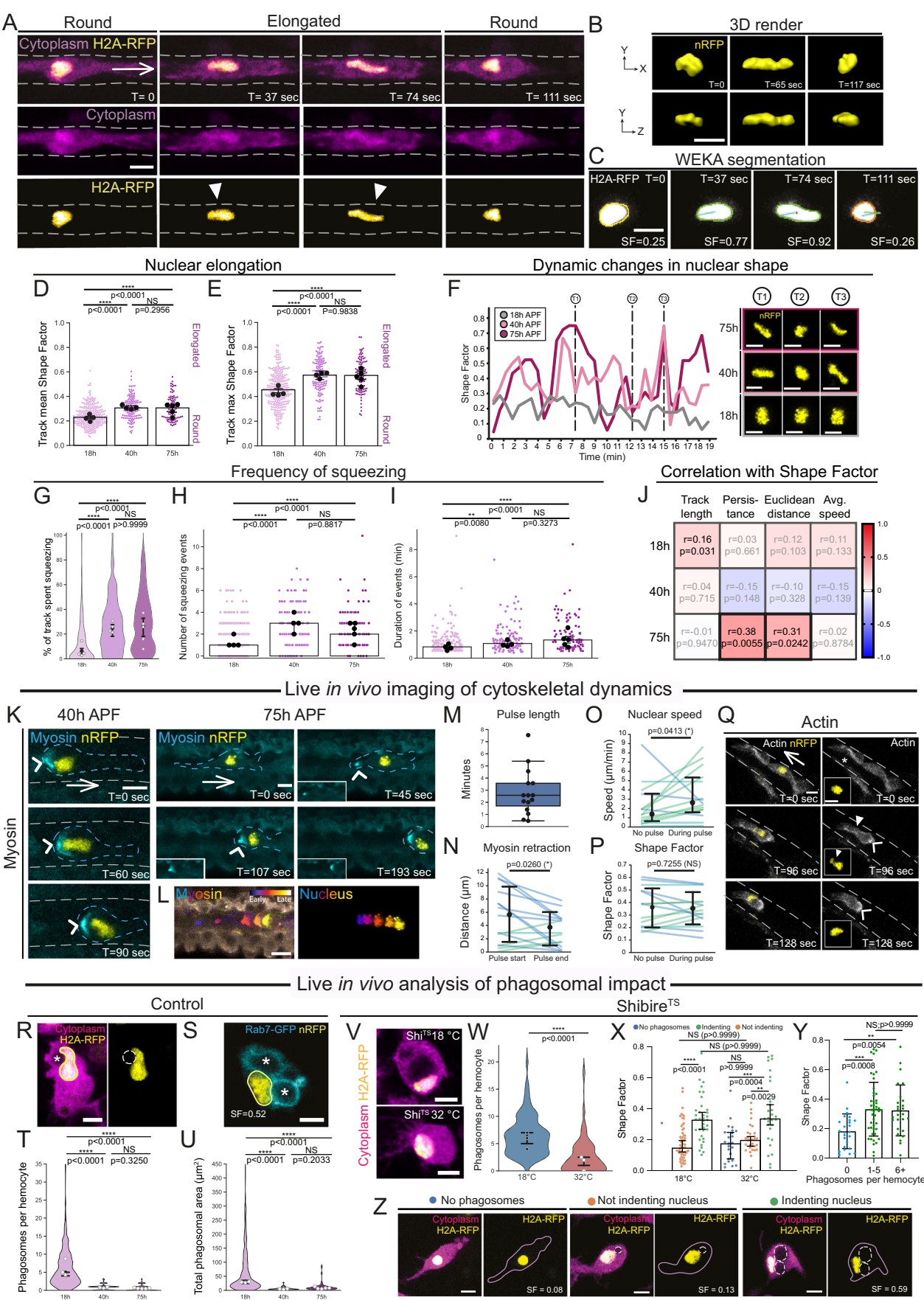

**Figure 2.** Immune cells exhibit striking nuclear dynamics during confinement in vivo.

(A) Timelapse of hemocyte migration (magenta cytoplasm, yellow nucleus) at 40 h APF; arrowheads indicate elongated nuclei. (B) 3D rendered example of nuclear elongation. (C) Nucleus in (A) detected via MIA plugin, with WEKA probability map (white); nuclear outline colour reflects the relative nuclear SF. (D, E) Scatter plots with small dots (individual nuclear tracks) and large black dots (wing medians). Bar chart shows median and 95% CI. The track average (D) and maximum (E) SF over 20 min at 18 h, 40 h and 75 h APF. (F) Example nuclear SF tracks over 20 min; example nuclei (yellow) correspond to measurements in the encircled timepoint shown on the graph. (G–I) Violin plot depicting proportion of the 20 min track nuclei spent squeezing (SF ≥ 0.4) (G). Scatter plots show the number (H) and length (I) of squeezing (SF ≥ 0.4) events, with bar showing median and 95% CI. (J) Correlation matrix of migration attributes with the average nuclear SF. R = Spearman r value, p = P values. Scale for r indicated on the colour bar. (K) Myosin II localisation (nucleus in yellow, Myosin II in cyan) at 40 h and 75 h APF, arrowheads label the rear-localised Myosin II signal and arrows indicate direction of migration. Cell body indicated with dashed line; related single channel Myosin II images shown in Fig. EV2M. (L) Time-projection of Myosin II and nuclear signal of 75 h APF hemocyte in (K) over 2.5 min. (M) Rear Myosin II pulse duration. (N) Distance of rear Myosin II signal from the nucleus, at the start and end of Myosin pulse. (O) Average nuclear migration speed, with and without a Myosin II pulse. (P) Average nuclear SF, with and without a Myosin II pulse. For (N–P), each line represents the linked data for a single cell; lines have been colour-coded (blue, negative gradient and green, positive gradient). (Q) Hemocyte Actin at 75 h APF (actin in grey, nucleus in yellow). Asterisk indicates leading edge signal, arrowhead rear signal. (R) 18 h APF hemocyte nucleus (cytoplasm in magenta, nucleus in yellow) deformed by a phagosome (asterisk). (S) 18 h APF hemocyte with nucleus (yellow) indented by Rab7-positive phagosomes (cyan, asterisk). (T, U) Number of phagosomes per hemocyte (T) and total phagosomal area per hemocyte, for those with phagosomes (U). (V–Z) Hemocytes expressing dominant-negative Shibire[ts1] (V, cytoplasm in magenta, nuclei in yellow) raised either in 18 °C or 32 °C, with number of phagosomes per cell (W) and nuclear SF (X and Y) stratified for temperature and/or phagosomal content, with representative hemocytes shown in (Z) (z-slices, cytoplasm in magenta, nucleus in yellow, phagosomes outlined). White arrows indicate direction of migration and grey dashed line marks approximate vessel wall location. Genotypes used were: Srp > GFP, SrpH2AmChe (A, C, R), Srp > nRFP (B, D–J), Srp > nRFP/Sqh^{AX3};sqh-Sqh-GFP (K–P), Srp > nRFP;Srp-GMA (Q), Srp > nRFP;UAS-Rab7.GFP (S), Srp > nRFP; Crq > GFP (T, U), Srp > GFP, SrpH2AmChe; UAS-Shi^{TS} (V–Z). Data information: All scale bars represent 5 μm. Violin plots (G, T, U, W) depict hemocytes, with wing medians in white dots and overall median and 95% CI shown in black. Boxplot with median and interquartile range (IQR, 25th to 75th percentile), with whiskers extending to the furthest data point within 1.5 times the IQR, individual values as black dots (M). Mean +/− SD (N, P) or median and 95% C.I. (O, X, Y) is shown in black. N = 4, 4 and 5 wings (D–J), N = 3, 3 and 5 wings (T, U) for 18 h, 40 h and 75 h APF. N = 3 wings and N = 15 hemocytes (M–P), N = 286, 139 and 102 tracks (D–H), N = 191, 124 and 95 tracks (I) and N = 182, 100 and 53 tracks (J) for 18 h, 40 h and 75 h APF. N = 216, 102 and 136 hemocytes (T), N = 206, 62 and 55 hemocytes (U) and 1308, 79 and 77 phagosomes (U) for 18 h, 40 h and 75 h APF. N = 3 and N = 5 wings with N = 96 and 56 hemocytes for 18 °C and 32 °C (W–X); N = 5 wings with N = 25, N = 47 and 26 hemocytes for (Y). One-way ANOVA (Kruskal-Wallis) with Dunn's multiple comparisons test (D–I, T, U), Paired t test (N, P), Wilcoxon matched-pairs signed rank test (O), or Mann-Whitney U test (W–Y) was used to calculate significance. Source data are available online for this figure.

range of SF values (Fig. EV2C). We calculated nuclear SF (and circularity) for nuclei with a variety of morphologies (Fig. EV2D) and defined a nucleus as undergoing deformation or 'squeezing' when the nuclear SF exceeded 0.4 (Fig. EV2D); comparison of nuclear SF with circularity measurements for identical nuclei (Fig. EV2D,E) suggested SF was a more sensitive measurement than circularity for quantifying nuclear deformation. Using this definition of squeezing, nuclei of leukocytes within wing vessels spent significantly more time 'squeezing' (Fig. 2G) than leukocytes migrating within the 18 h APF wings. Further inspection of nuclear squeezing revealed that nuclei of leukocytes within vessels undergo 'squeezing' events more frequently (Figs. 2H and EV2F), and each squeezing event lasts longer than those of 18 h APF leukocytes (Fig. 2I). Correlative analysis of leukocyte behaviour revealed that the nuclear shape of leukocytes migrating within the narrow convoluted 75 h APF vessels was highly correlated with the persistence of leukocyte migration and the Euclidean distance travelled (Figs. 2J and EV2G), suggesting that leukocytes with increased nuclear deformation migrate more persistently and can cover greater distances. Increased nuclear deformation of leukocytes might thus be key to supporting effective navigation within these more confining environments. To investigate whether the observed nuclear dynamics were associated with vessel formation (or rather reflected a change occurring with developmental time), we compared nuclear dynamics in vessels of control wings with those in which vessel formation had been inhibited (using epithelial-specific inhibition of β-integrin via mys-RNAi, Fig. EV2H–J); leukocyte nuclei exhibited increased nuclear dynamics when migrating within the more confining vessel environment of wings of equivalent developmental age (Fig. EV2J).

Recent in vitro evidence suggests that the dynamic interaction of the nucleus with the perinuclear cytoskeleton might aid propulsion to squeeze migrating cells through artificial confinements (Keys

et al, 2024; Thiam et al, 2016; Lomakin et al, 2020; Venturini et al, 2020). Whether a conserved mechanism aids confined in vivo migration remains debated; while collectively migrating epithelial-like ('border') cells in the Drosophila ovary exhibit actomyosin contractions during confined migration (Penfield and Montell, 2023), Drosophila embryonic macrophages instead possess a protective actomyosin cage that in fact 'shields' the nucleus, preventing deformation (Belyaeva et al, 2022). Here, our live-imaging of endogenous GFP-tagged non-muscle Myosin II (Fig. 2K and Movie EV5) revealed that mature leukocytes migrating within narrow channels in vivo exhibited substantial rear waves of Myosin II that moved rapidly forwards from the posterior cortex towards the nucleus over the course of 2–3 min (Fig. 2K); this dynamic Myosin II wave culminated in nuclear propulsion forwards ('nucleokinesis') as the Myosin II pulse reached the nuclear periphery (Figs. 2K–O and EV2K–M). Rear-localised actomyosin contraction was associated with more rapid forwards nuclear movement (Fig. 2O), although the Myosin II pulse did not correlate with the extent of nuclear squeezing, suggesting that additional mechanisms might impact nuclear deformation (Fig. 2P). Live-imaging of leukocyte actin dynamics revealed similar behaviour to that of Myosin II, with initial localisation at the leading edge and a subsequent rear wave of actin coinciding with posterior retraction (Fig. 2Q). Actin could also be seen to localise around the nuclear periphery during deformation (Fig. EV2N, arrowheads), which may be akin to the Arp2/3-dependent actin polymerisation promoting nuclear deformation of dendritic cells during in vitro confinement (Thiam et al, 2016). Our previous work revealed that integrin-based adhesion is not essential for leukocyte movement within the wing vessels, suggesting that Drosophila leukocytes do not rely on adhesion to the vessel substratum (Thuma et al, 2018). Our data thus suggest leukocytes move rapidly within these narrow channels in vivo using a 'pseudopodal amoeboid style' (Lämmermann and

Sixt, 2009), with minimal reliance on cell-substrate adhesions, but actin polymerisation at the cell front and actomyosin contractility at the rear cortex to propel the nucleus forwards. This mirrors that of mammalian neutrophils in vitro, which in fabricated 3D environments move in a low-adhesive and largely integrin-independent manner with myosin-II dependent contraction at the cell rear (Malawista and De Boisfleury Chevance, 1997; Lämmermann et al, 2008; Lämmermann and Sixt, 2009).

Intriguingly, despite the relatively unconstrained environment for leukocytes within 18 h APF wings, their nuclei were often surprisingly deformed (Fig. 2D,E), albeit at significantly lower frequency than leukocytes within vessels. Leukocytes within 18 h APF wings frequently contained multiple, large phagocytic vesicles (visualised using Rab7-GFP) that often contacted and appeared to indent the nuclear envelope (Fig. 2R–U); in contrast, leukocytes constrained within vessels contained significantly fewer phago-somes (Figs. 2R–U and EV2O–Q). These data, along with quantification (Fig. EV2R), suggest that nuclear deformation observed in leukocytes of 18 h APF wings could derive from intracellular phagosomal pressure. To test this hypothesis, we suppressed leukocyte phagocytosis (by expressing the temperature-sensitive dominant negative Dynamin GTPase *shibire^{ts1}* (Vicario et al, 2001) specifically in leukocytes, Fig. 2V–Z). Leukocytes expressing *shibire^{ts1}* within 18 h APF pupae (raised at the restrictive temperature, 32 °C, to activate the construct) contained significantly fewer phagosomes than those in controls (raised at the permissive temperature, 18 °C) (Figs. 2V,W and EV2S). Analysis of nuclear shape revealed that nuclear deformation was significantly greater for leukocytes in which phagosomes pushed against the nuclear envelope, compared to those leukocytes that lacked phagosomes or possessed phagosomes minimally contacting the nucleus (Fig. 2X); nuclear shape factor did not differ for leukocytes containing phagosomes whether the pupae were raised at 18 °C or shifted to 32 °C (Fig. 2X). While nuclear deformation was significantly increased in leukocytes containing up to five phagosomes (compared to leukocytes lacking phagosomes), the extent of nuclear deformation did not increase further as leukocytes accumulated additional phagosomes (Fig. 2Y). Intriguingly, a similar phenomenon could occur in mammalian cells, as recent work demonstrated cytoplasmic lipid droplets can indent the macrophage nucleus in vitro (Ivanovska et al, 2023). Our data suggest that migrating cells, particularly highly phagocytic leukocytes, may endure nuclear deformation even in environments lacking external confinement due to the ability of cytoplasmic organelles to exert internal pressure on the nucleus.

## Leukocytes adapt their nuclear lamina composition upon confinement in vivo

Nuclear mechanical properties are largely determined by a cell's unique nuclear architecture, including the composition of the nucleoskeleton (nuclear lamina and components linking it to the nuclear envelope, NE) and the nuclear interior (chromatin). While there are marked differences in nuclear stiffness between diverse cell types, individual cells could also change their nuclear stiffness during their lifetime as they differentiate or move into new environments (Shin et al, 2013; Swift et al, 2013). Indeed, embryonic stem cells lose their physical (and transcriptional)

plasticity during development due to increased expression of NE proteins and modified chromatin structure (Pajerowski et al, 2007).

The nuclear lamina, which underlies the NE, functions as a structural and mechanical scaffold for the nucleus (Gruenbaum and Foisner, 2015). The lamina is composed of a dense meshwork of intermediate filament proteins called Lamins (A-type and B-type) that assemble into structured networks with distinct properties. While at least one B-type Lamin is constitutively expressed in all cells, expression of A-type Lamins is developmentally regulated (Broers et al, 1997). A-type Lamins (Lamins A and C) are considered the main contributors to nuclear stiffness and provide key structural support to the nuclear envelope (Swift et al, 2013; Niethammer, 2021; Lammerding et al, 2006). B-type Lamins (Lamin B1 and Lamin B2) contribute to the elasticity of the nucleus and help anchor the nucleus to the cytoskeleton (Harada et al, 2014). Nuclear rigidity is thus largely determined by the relative levels (stoichiometry) of A- and B-type Lamins (Swift et al, 2013). As in mammals, *Drosophila* possess two Lamin subtypes (Muñoz-Alarcón et al, 2007), with a single B-type Lamin (*Drosophila* Lamin Dm0, dLamB) and a single A-type Lamin (*Drosophila* Lamin C, dLamC) (Fig. 3A). Live-imaging of the *Drosophila* B-type Lamin (either via RFP-tagging of endogenous dLamB, Fig. 3B or leukocyte-specific expression of GFP-tagged dLamB, Fig. 3C) enabled us to follow nuclear envelope dynamics during leukocyte nuclear deformation at high spatio-temporal resolution in vivo (Movie EV6). Live-imaging revealed similar dynamic lamina deformations as that observed with nuclear RFP (Fig. 3B,C) as leukocytes migrated within pupal wings (Fig. 3B,C). In contrast, live-imaging of the *Drosophila* A-type Lamin (via GFP-tagging of endogenous dLamC, Fig. 3D) suggested that leukocytes (Fig. 3D, arrow) possessed much lower A-type Lamin levels than the adjacent (relatively immobile) epithelial cells that exhibited a clear NE-localised dLamC signal.

We envision that cells could dynamically change the composition of their nuclear lamina during migration to modify mechanical properties of the nucleus, enabling rapid cellular adaptation to external mechanical cues. This could enable migrating cells to match their nuclear plasticity to their ever-changing local environment, beyond any earlier developmental changes that may have already softened the nuclear lamina during differentiation (Olins et al, 2008; Rowat et al, 2013); indeed, naive T lymphocytes upregulate Lamin A/C expression following activation to ensure they remain stationary during engagement with an antigen presenting cell (González-Granado et al, 2014).

To explore whether *Drosophila* leukocytes dynamically adapted their nuclear lamina composition in vivo, we comprehensively quantified leukocyte Lamin levels, using immunohistochemistry on ex vivo leukocytes (Fig. 3E–I) as well as in vivo whole-mount tissues (Fig. 3J–M). While both Lamin subtypes could be detected in the leukocyte nuclear envelope at 18 h, 40 h and 75 h APF (Figs. 3F and EV3A), the relative levels of each Lamin subtype changed as leukocytes transitioned into the narrow channels (Fig. 3G–I). Leukocytes maintained consistently low levels of dLamC as they transitioned into vessels (Fig. 3G), with a small transient increase in dLamC in 40 h APF leukocytes that was not sustained in leukocytes within 75 h APF vessels. In contrast, dLamB levels significantly increased within the leukocyte nuclear envelope as leukocytes entered the more constricted vessel environment (Fig. 3H). Leukocytes within 18 h and 75 h wings possessed similar

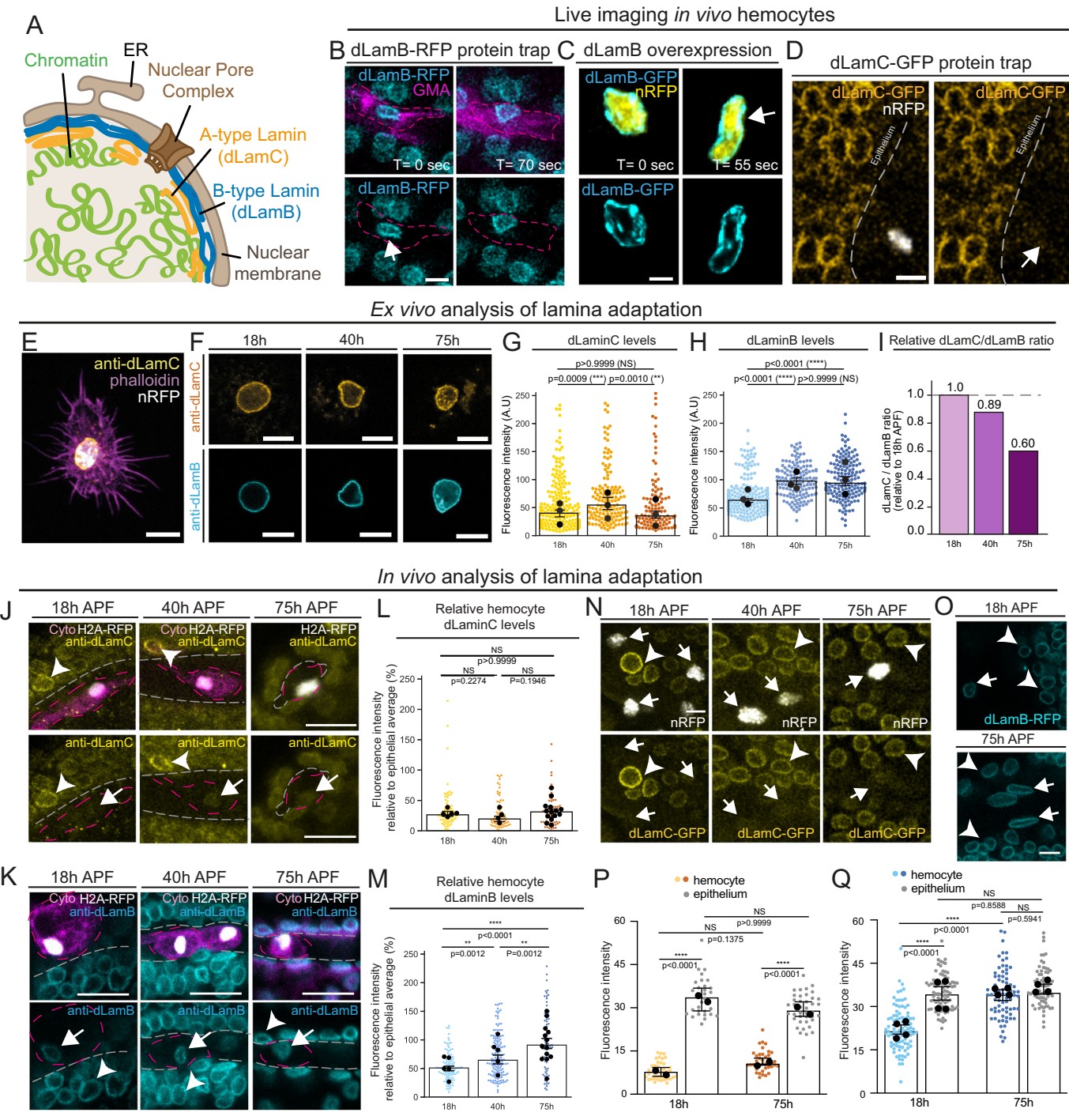

histone intensities (Fig. EV3E,F) and appeared at stage G0/G1 of the cell cycle (Fig. EV3G–I), suggesting the observed increase in dLamB was not associated with elevated leukocyte DNA content (which has been previously linked with transition from G0/G1 to G2/M (Vashisth et al, 2021)). These data suggest that relative Lamin stoichiometry progressively changes within leukocytes as they transition into new, more confined environments in vivo, with levels of dLamB increasing relative to dLamC, suggesting progressive softening of leukocyte nuclei (Fig. 3I). To assess

whether this lamina modulation was specific to leukocytes within the wing or rather reflected a general developmental change (in leukocytes elsewhere in the pupae), we analysed dLamB levels in wing leukocytes and those isolated from the main pupal body (Fig. EV3J–L); these data suggest dLamB levels increase specifically in wing leukocytes from 18 h to 75 h APF (Fig. EV3J,K).

To probe whether these dynamic changes in lamina composition were specific to leukocytes or common to other wing (e.g. epithelial) tissues, we assessed Lamin levels in leukocytes and

◄ **Figure 3.  Immune cells modify their nuclear lamina composition upon transition to vessels.**

(A) Schematic of key nuclear structures, with *Drosophila* Lamin nomenclature. (B) Hemocyte (magenta, outlined) with endogenous dLamB-RFP (cyan) in a 75 h APF vessel in vivo. Arrow indicates squeezing nucleus. (C) Hemocyte nucleus (yellow) over-expressing dLamB-GFP (cyan) at 18 h APF in vivo; arrow indicates dynamic lamina rearrangements during nuclear shape changes. (D) Hemocyte nucleus (white) with endogenous dLamC-GFP (yellow) at 18 h APF in vivo; arrow indicates lack of dLamC in the hemocyte nucleus. (E) Representative ex vivo hemocyte (nucleus in white, anti-dLamC in yellow, Actin in magenta) extracted from a 40 h APF pupal wing. (F) Ex vivo hemocyte nuclei stained with anti-dLamC (yellow) or anti-dLamB (cyan) from 18 h, 40 h and 75 h APF pupal wings; single z-stacks are shown. (G, H) Quantification of anti-dLamC (G) and anti-dLamB (H) levels in ex vivo hemocytes at 18 h, 40 h and 75 h APF. (I) Relative dLamC/dLamB ratio of ex vivo hemocyte Lamin levels when normalised to 18 h APF ratio. (J, K) In vivo hemocytes (cytoplasm in magenta, outlined, nucleus in white) stained with anti-dLamC (J) or anti-dLamB (K) at 18 h, 40 h and 75 h APF. Epithelial border is marked with grey dashed line; arrowheads label epithelial nuclei and arrows label hemocyte nuclei. (L, M) Relative hemocyte nuclear anti-dLamC (L) or anti-dLamB (M) fluorescence intensity when normalised to epithelial nuclear average within the same age group. (N–Q) Live-imaging of endogenous dLamC-GFP (N, yellow; arrowheads indicate epithelial dLamC-GFP and arrows indicate lack of dLamC in hemocyte nuclei) or endogenous dLamB-RFP (O, cyan; arrowheads and arrows indicate dLamB in epithelial or hemocyte nuclei, respectively) in vivo; quantification of endogenous dLamC-GFP (P) or dLamB-RFP (Q) in hemocyte and adjacent epithelial cells. Genotypes used were: *dLamB-tagRFP;SrpGMA* (B), *Srp > nRFP;UAS-dLamB-GFP* (C), *Srp > nRFP;dLamC-tagGFP* (D, N, P), *Srp > nRFP* (E–I), *Srp > GFP, SrpH2AmChe* (J–M) and *dLamB-tagRFP* (O, Q). Data information: Scale bars represent 5 μm (B, D–F, N, O), 2 μm (C) or 10 μm (J, K). Scatter plots with small dots (hemocytes) and large black dots (medians of technical replicates (G, H) or wing sections (L, M)). Bar chart shows median and 95% CI. $N = 3$ technical replicates with >8 wings each (G, H), and $N = 5$, 4 and 14 wing sections (L) and $N = 5$, 5 and 13 wing sections (M) from >8 wings (separate pupae) stained per age. $N = 211$, 148 and 125 (G), 215, 150 and 161 (H), $N = 74$, 67 and 72 nuclei (L), $N = 106$, 151 and 84 nuclei (M) measured for 18 h, 40 h and 75 h APF, respectively; $N = 41$ and 40 hemocyte nuclei and $N = 30$ and 40 epithelial nuclei (P) or $N = 97$ and $N = 77$ hemocyte nuclei and $N = 86$ and 77 epithelial nuclei (Q) measured for 18 h and 75 h APF, respectively. One-way ANOVA (Kruskal-Wallis) with Dunn's multiple comparisons test to calculate significance (G, H, L, M, P, Q). Source data are available online for this figure.

adjacent epithelial cells in their in vivo tissue context (Figs. 3J–Q and EV3B–D,M,N). We performed both immunostaining of wing tissue (Fig. 3J–M) and live-imaging of endogenously tagged Lamins (Fig. 3N–Q). To mitigate for different immunostaining efficiencies across developmental stages, in vivo immunostaining was utilised to compare the relative levels of Lamins between leukocytes and epithelia at each stage (Fig. 3J–M); these data were complemented by live-imaging of endogenous Lamins for comparison of absolute levels of Lamins in leukocytes and epithelia (Fig. 3N–Q). Consistent with earlier live-imaging of endogenously tagged dLamC (Fig. 3D), leukocytes in vivo had significantly less dLamC than neighbouring epithelial cells (~25%), a trend that was uniformly observed across all wing environments (Figs. 3J,L,N,P and EV3B,M). Thus, despite dLamC being detected in isolated leukocytes ex vivo (Fig. 3F,G), minimal dLamC could be detected by live-imaging of endogenously tagged dLamC (Fig. 3D,N,P) or in vivo immunostaining (Fig. 3J). As A-type Lamins can hinder confined migration (Harada et al, 2014), wing leukocytes may maintain consistently low Lamin-A levels to facilitate nuclear plasticity. In contrast, leukocytes possessed more dLamB (~50% of that observed in neighbouring epithelial cells) even in unconfined environments and this increased significantly as leukocytes transitioned into vessels (Figs. 3K–M,O,Q and EV3C,N). This is consistent with studies on the lamina composition of human hematopoietic cells, where the expression of B-type Lamins is highly variable across cell types (by up to 30-fold) but Lamin-A varies far less (Shin et al, 2013); moreover, non-circulating human cells (e.g. marrow-resident mesenchymal stromal cells, MSCs) exhibit much higher Lamin-A levels than their circulating counterparts (e.g. granulocyte/monocyte lineages) (Shin et al, 2013).

Ultimately, our data suggest that leukocytes migrating within the constrained vessel environments in vivo might dynamically adapt their nuclear mechanics and fine-tune their nuclear lamina composition by altering the stoichiometry of A- and B-type lamins, with higher levels of the elastic B-type dLamB than the 'stiffer' A-type dLamC. Since experimental reduction of Lamin-A in neutrophils and monocytes increases their net migration in vitro through small capillary-sized micropores (Shin et al, 2013), such dynamic adaptation in vivo might increase ('prime') the cell's ability to deform its nucleus to aid more effective movement through the confined tissue.

## Leukocyte lamina adaptation is required for nuclear plasticity and effective immunosurveillance upon transition to confined environments in vivo

To investigate the role of this in vivo lamina adaptation, we integrated live in vivo imaging with leukocyte-specific genetic perturbation of each Lamin subtype (Fig. 4). Given the increase in nuclear dLamB as cells transitioned to moving within narrow channels, we investigated whether elevated dLamB was required to support leukocyte nuclear dynamics in vivo. Leukocyte-specific downregulation of *dLamB* (via srp-Gal4 driven *UAS-RNAi*) reduced dLamB levels within vessel-localised leukocytes to that normally observed within unconfined control leukocytes (at 18 h APF), without affecting leukocyte levels of dLamC (Appendix Fig. S1A–D); this enabled us to investigate the role of the confinement-associated increase in leukocyte dLamB. Leukocytes with reduced dLamB exhibited perturbed nuclear dynamics across all wing environments, with most dramatic effects for leukocytes migrating within the constrained vessel environments. *dLamB-RNAi* leukocytes within vessels were rounder (Fig. 4A,B; Appendix Fig. S1E) and exhibited significantly reduced frequency of nuclear deformation compared to their age-matched controls (Fig. 4C,D; Appendix Fig. S1F), behaving remarkably similar to control leukocytes within the unconstrained 18 h APF environment. Intriguingly *dLamB-RNAi* nuclei were significantly smaller than those of controls at 40 h APF (Appendix Fig. S1G). Nuclear deformation phenotypes could be replicated using a second independent *dLamB-RNAi* line (termed "*dLamB-RNAi 2*", Appendix Fig. S1H–J). These data suggest that confinement-associated increases in leukocyte dLamB are normally required to support the heightened nuclear plasticity during wing vessel migration.

We next explored whether leukocytes require sustained low levels of dLamC for their precise nuclear dynamics in vivo. Leukocyte-specific inhibition of *dLamC* (via srp-Gal4 driven RNAi) reduced dLamC levels, whereas leukocyte-specific over-expression

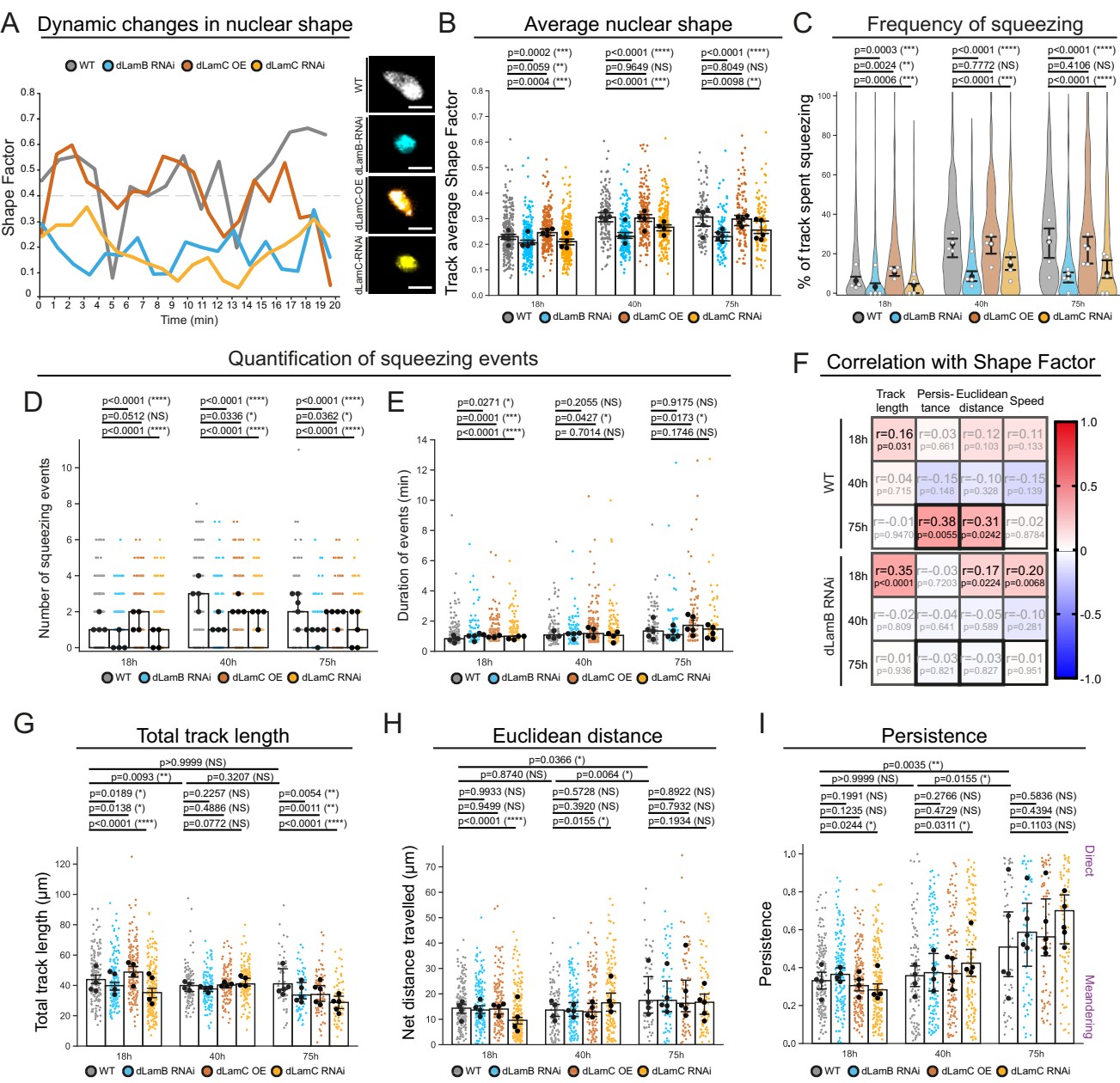

**Figure 4. Immune cells finely-tuned nuclear lamina composition supports nuclear plasticity in vivo.**

(**A**) Example nuclear SF dynamics from 20 min migration tracks at 75 h APF. (**B**) The track average SF measured over 20 min imaging periods at 18 h, 40 h and 75 h APF following hemocyte-specific Lamin manipulation. (**C**) Violin plot depicting the percentage of nuclear tracks during which SF is ≥0.4 at 18 h, 40 h and 75 h APF. Wing medians (white dots), with sample median and 95% CI (black). (**D, E**) The number (**D**) and length (**E**) of SF peaks (SF ≥ 0.4) during a 20 min imaging period. An outlier (at 18.1 min) for *dLamB-RNAi* at 18 h APF was excluded from graph in (**E**) but included in data analysis. (**F**) Correlation matrix of migration attributes and average nuclear SF in *control* (WT) and *dLamB-RNAi* hemocytes. R = Spearman r value, *p* = P values; scale for r indicated on the colour bar. (**G–I**) Hemocyte migration track length (**G**), Euclidean distance (**H**) and persistence (**I**) measured over 15 min periods. Genotypes used were: *Srp > nRFP* ('WT' controls, **A–I**), *Srp > nRFP;UAS-dLamB-RNAi* (**A–I**), *Srp > nRFP;UAS-dLamC-OE* (**A–E, G–I**) and *Srp > nRFP;UAS-dLamC-RNAi* (**A–E, G–I**). Data information: Scale bars represent 5 µm (**A**). Scatter plots with small dots (individual nuclear tracks) and large black dots (wing medians). Bar chart shows median and 95% CI (**B, D, E, G–I**). N = 4, 4 and 5 wings for 18 h, 40 h and 75 h APF, respectively (**B–I**). N = 286, 139 and 102 hemocytes (WT); 237, 183 and 96 hemocytes (*dLamB-RNAi*); 238, 193 and 113 hemocytes (*dLamC-OE*); 293, 199 and 101 hemocytes (*dLamC-RNAi*) (**B–D**). N = 182, 100 and 53 tracks (WT); 177, 121 and 53 (*dLamB-RNAi*) (**F**). N = 191, 124 and 95 tracks (WT); 121, 134 and 63 (*dLamB-RNAi*); 182, 175 and 95 (*dLamC-OE*); 154, 158 and 66 (*dLamC-RNAi*) (**E**). N = 190, 103 and 55 hemocytes (WT); 178, 127 and 59 (*dLamB-RNAi*); 172, 131 and 77 (*dLamC-OE*); 223, 142 and 74 (*dLamC-RNAi*) (**G–I**). One-way ANOVA (Kruskal-Wallis) with Dunn's multiple comparisons test used for significance between control hemocytes (**G–I**), Mann-Whitney U test used to calculate significance for *dLamB-RNAi, dLamC-OE* and *dLamC-RNAi* (**B–E, G–I**). Source data are available online for this figure.

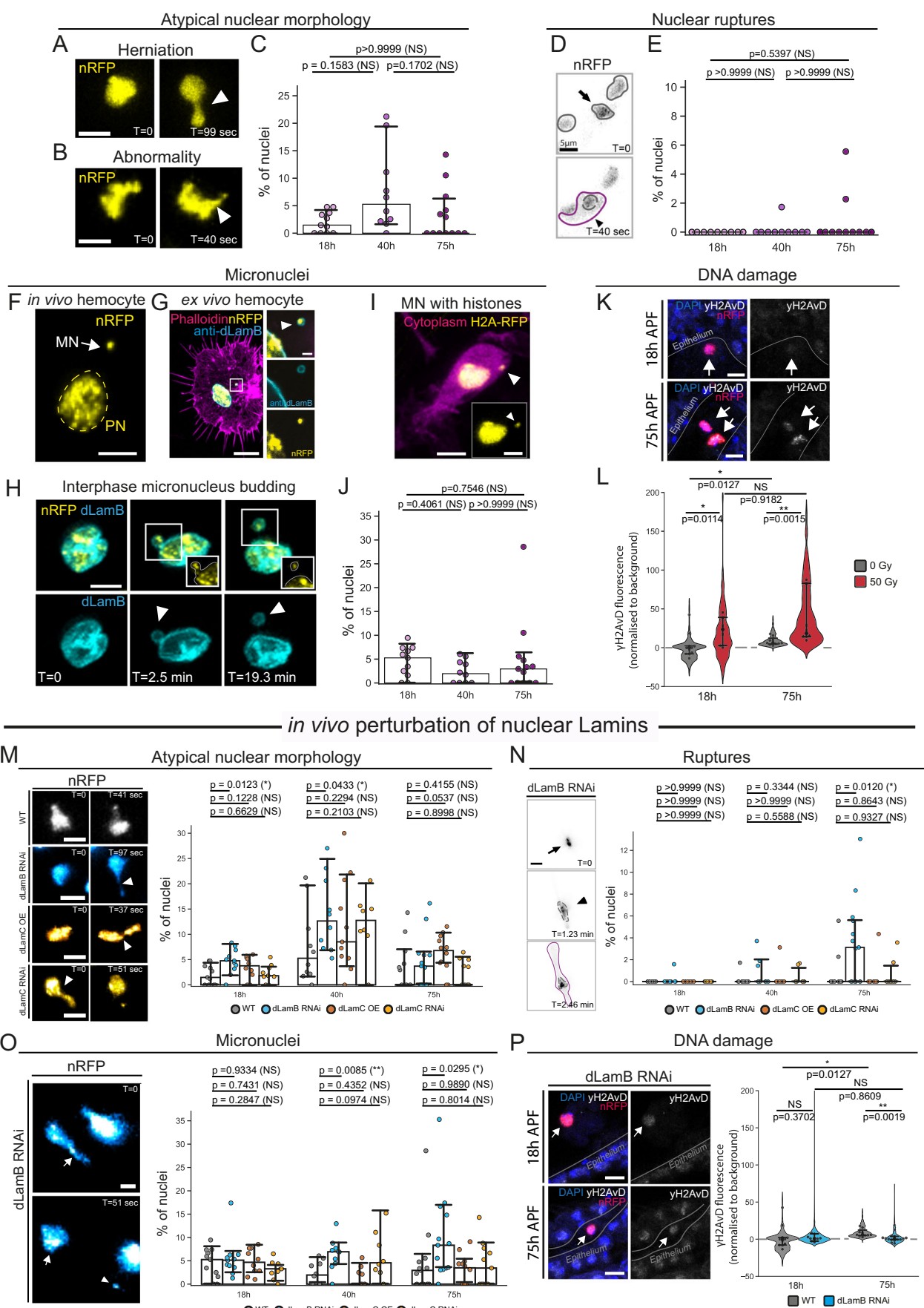

**Figure 5. Precisely adapted lamina composition offers nuclear protection during confined vessel migration in vivo.**

(A, B) Hemocyte nuclei with nuclear instability including herniations (A, arrowhead) or with abnormal morphology (B, arrowhead). (C) Percentage of nuclei with atypical morphologies over 20 min imaging period. (D) 75 h APF hemocyte before rupturing (arrow) and the release of nuclear RFP into the cytoplasm after rupture (arrowhead, hemocyte outlined in magenta). Nuclei are outlined in grey. (E) Percentages of ruptured nuclei. (F) In vivo hemocyte primary nucleus (PN, yellow) with a micronucleus (MN, arrow). (G) MN (arrowhead) in 40 h APF ex vivo hemocyte (nucleus in yellow, anti-dLamB in cyan and Actin in magenta). (H) MN (arrowhead) budding from in vivo hemocyte nucleus (nucleus in yellow, dLamB-GFP in cyan). (I) Histones within MN (arrowhead) of 18 h APF hemocyte (histones in yellow, cell body in magenta). (J) Percentage of nuclei with MN. (K) Cryosections of 18 h and 75 h APF pupal wings stained with DAPI (blue) and anti-γH2AvD (grey). Hemocyte nuclei are in red (arrows). (L) Hemocyte nuclear γH2AvD fluorescence normalised to background before (0 Gy) or after irradiation (50 Gy). (M) Example images of atypical nuclear morphologies (arrowheads) in *control* (WT), *dLamB-RNAi*, *dLamC-OE* and *dLamC-RNAi* hemocyte nuclei, with quantification of percentage of nuclei with atypical nuclear morphology. (N) *dLamB-RNAi* hemocyte before (arrow) and after rupturing (arrowhead, hemocyte outlined in magenta). (O) Example of deformation-associated MN formation in *dLamB-RNAi* hemocytes, arrowhead labels MN budding after nuclear elongation (arrows). (P) Cryosections of *dLamB-RNAi* 18 h and 75 h APF pupal wings stained with DAPI (blue) and anti-γH2AvD (grey). Hemocyte nuclei are in red (arrows). Violin plots show *control* (WT) and *dLamB-RNAi* hemocyte nuclear γH2AvD fluorescence after normalisation to background. Genotypes used were: *Srp > nRFP* (A–E, F, G, J–P), *Srp > nRFP; UAS-dLamB-GFP* (H), *Srp > GFP, SrpH2AmChe* (I), *Srp > nRFP; UAS-dLamB-RNAi* (M–P), *Srp > nRFP; UAS-dLamC-OE* (M–O) and *Srp > nRFP; UAS-dLamC-RNAi* (M–O). Data information: Scale bars represent 5 µm (A, B, F, H, I, K, M, O, P), 10 µm (G) and 1 µm (G, insets). Large dots in scatter plots show percentages per wing, bar chart shows median and 95% CI of all wings (C, E, J, M–O). Violin plots show all nuclei, sample medians shown as dots and 95% CI shown in black (L, P). N = 11, 10 and 13 wings for WT; 11, 10 and 13 wings for *dLamB-RNAi*; 10, 11 and 12 wings for *dLamC-OE*; 9, 8 and 11 wings for *dLamC-RNAi* (C, E, J, M–O). N = 616, 382 and 209 hemocytes for WT; 791, 444 and 343 for *dLamB-RNAi*; 540, 484 and 338 for *dLamC-OE*; 567, 402 and 307 for *dLamC-RNAi* (C, E, J, M–O). N = > 450 nuclei per condition, 3 technical replicates with >4 wings for each except for WT 0 Gy γH2AvD with 2 technical replicates (L, P). One-way ANOVA (Kruskal-Wallis) with Dunn's multiple comparisons test (C, E, J) or Mann-Whitney U test (L–P) was used to calculate significance. Source data are available online for this figure.

of *dLamC* elevated nuclear lamina levels of dLamC (Appendix Fig. S1A). Despite control leukocytes possessing minimal dLamC, *dLamC-RNAi* perturbed nuclear dynamics in vivo, particularly for leukocytes migrating under vessel confinement (Fig. 4A–E) with reduced overall nuclear deformation, dynamics and frequency of squeezing—although to a lesser extent than that caused by *dLamB-RNAi*. Elevation of leukocyte dLamC had surprisingly less effect on nuclear deformation in vivo (Fig. 4A–E)—we hypothesise this is because experimental elevation of leukocyte dLamC was also accompanied by increased levels of dLamB (Appendix Fig. S1B). These data suggest that leukocytes in vivo possess the remarkable ability to compensate for aberrantly high dLamC levels (that could otherwise increase nuclear stiffness) by dynamically elevating their expression of dLamB, to maintain the relative dLamC/dLamB ratio nearer that of controls (Appendix Fig. S1C) and counteract potential defects.

Given that perturbation of the nuclear lamina composition disrupted nuclear deformation dynamics, we explored whether this had downstream consequences for leukocyte physiological function (e.g. migratory behaviour, Fig. 4F–I) in vivo. Since the nuclear lamina mechanically links the nuclear envelope to the cellular cytoskeleton, loss of dLamB might hinder nuclear (and cellular) dynamics by disconnecting Myosin II contractility from the nucleus. Consistent with this, the previously observed strong correlation between nuclear deformation (SF) and key aspects of leukocyte migration (persistence or Euclidean distance travelled) for control leukocytes migrating within the narrow 75 h APF vessels was abolished following *dLamB-RNAi* (Fig. 4F). Within the convoluted narrow vessels of 75 h APF wings, *dLamB-RNAi* leukocytes (with dLamB levels resembling those of control 18 h APF leukocytes) exhibited significantly reduced migratory capacity, travelling smaller total distances than age-matched controls (Fig. 4G; Appendix Fig. S1K); similar migratory defects for leukocytes within the narrow 75 h vessels were observed following *dLamC-RNAi*. These data suggest that loss of nuclear plasticity (or nuclear-cytoskeletal connection) disrupted the ability of leukocytes to navigate rapidly in complex tissues and perform key roles in immune surveillance.

## A precisely adapted lamina composition offers nuclear protection during confined vessel migration in vivo

Successful immune surveillance through confined environments likely not only requires fine-tuning of nuclear mechanics to enable sufficient nuclear deformation for migration through narrow pores, but the lamina must also offer sufficient protection to survive large nuclear envelope distortions (and limit catastrophic effects on nuclear health and cell survival). Indeed, in vitro studies suggest that while nuclear lamina stiffness can impede 3D migration, nuclear softness might also limit survival (Harada et al, 2014); this may explain why mature neutrophils with a soft malleable nucleus die within days of differentiation (Pillay et al, 2010; Raab et al, 2016).

We explored whether *Drosophila* leukocytes exhibited nuclear instability during confined in vivo migration and whether the nuclear lamina adaptations conferred important nuclear protection. Using live time-lapse imaging of control leukocyte nuclear dynamics, we could occasionally detect nuclear 'herniations' (Fig. 5A and Movie EV7) or other nuclear abnormalities (e.g. locally pinched nuclear regions, Fig. 5B), which occurred more frequently as leukocytes first transitioned into vessels (Fig. 5C; Appendix Fig. S2A,B). Rare nuclear ruptures were also observed within control leukocytes, but only for those migrating within vessels in vivo (Fig. 5D,E, transient escape of nuclear RFP from the nucleus into the cytoplasm). Surprisingly, control leukocytes in vivo occasionally contained micronuclei (Fig. 5F–J). Micronuclei (MN) are small cytoplasmic structures, originating from the main nucleus, that contain chromatin (Guo et al, 2019). Until now, MN have been observed predominantly within cancer cells (Vázquez-Diez et al, 2016) and their formation has rarely been captured live within in vivo settings, particularly in non-dividing cells. MN are thought to form during cell division but have more recently been linked to nuclear envelope rupture (Guo et al, 2019); DNA damage accumulates in MN of cancer cells and could contribute to their characteristic genomic rearrangements (Hatch et al, 2013). Here, similar to the main nucleus, leukocyte MN were enveloped by nuclear lamina (Fig. 5G) and contained chromatin

(Fig. 5I). Through in vivo live-imaging, we observed the budding of MN directly from the nucleus of non-dividing leukocytes (Fig. 5H and Movie EV7); MN punctae persisted in the leukocyte cytosol during further cell migration (for the remainder of imaging) and were observed in leukocytes across all wing environments (Fig. 5J).

Analysis of DNA damage within leukocytes in vivo (via immunostaining for γH2AvD, the *Drosophila* γ-H2AX equivalent) revealed that DNA damage levels were minimal in nuclei of 18 h APF leukocytes, but significantly increased in nuclei of leukocytes migrating under confinement (Fig. 5K,L; Appendix Fig. S2C); this parallels the increased DNA damage within immune and cancer cells migrating through artificial micron-scale constrictions in vitro (Pragya Shah et al, 2021; Raab et al, 2016). The specificity and reliability of γH2AvD staining was confirmed by exposing tissues to 50 Gy irradiation (for 2 h), which significantly elevated leukocyte DNA damage at both pupal stages (Fig. 5L). Nevertheless, while immune cells migrating under physiological confinement in vivo experience elevated DNA damage, they are largely able to resist more dramatic nuclear instability events (e.g. nuclear ruptures) despite exhibiting frequent cycles of nuclear deformation. This is in contrast to cancer cells, where mechanical deformations frequently induce nuclear rupture in vitro (Denais et al, 2016; Raab et al, 2016; Xia et al, 2018).

We envision the dynamic lamina adaptation in leukocytes upon transition to narrow, vessel-like channels could facilitate nuclear resilience during in vivo confinement. We thus explored whether the leukocyte's precisely adapted lamina composition was also required for nuclear health during physiological confinement in vivo (Fig. 5M–P; Appendix Fig. S2D–I). *dLamB-RNAi* leukocytes (with dLamB levels more comparable to controls at 18 h APF) experienced significantly more nuclear instability events during in vivo migration (Fig. 5M–O and Movie EV8). Crucially, the type of abnormalities observed in *dLamB-RNAi* leukocytes depended on the specific wing environment. When first transitioning to 40 h APF wing vessels, leukocytes with low dLamB were significantly more likely to exhibit nuclear atypicalities (Fig. 5M and Movie EV8) and MN formation (Fig. 5O). MN were regularly observed to form following nuclear deformation in leukocytes lacking dLamB (Movie EV8) with MN budding from the distorted nuclear region, suggesting that elevated dLamB normally enhances the resilience of the nuclear lamina to enable reversible nuclear deformation without MN formation.

As leukocytes with abnormally low dLamB transitioned into the narrow, convoluted 75 h APF vessels, they were significantly more likely to experience nuclear ruptures (Fig. 5N). These data suggest that reduction of dLamB within the leukocyte nuclear lamina might have distinct consequences for the nucleus depending on the specific microenvironment. Frequent herniations (or uncontrolled deformations) experienced by dLamB-deficient leukocytes in vessels from 40 h APF onwards might progressively weaken the nucleus and ultimately lead to nuclear envelope rupture in the most confining microenvironments (at 75 h APF). Surprisingly, despite the instability of dLamB-deficient nuclei, these nuclei exhibited less DNA damage than age-matched controls at 75 h APF (Fig. 5P; Appendix Fig. S2F–I); perhaps the lack of nuclear deformation in these cells makes them less prone to accumulating genomic damage. Given that nuclear rupture can release DNA repair factors and compromise cellular DNA repair capacity (Ivanovska et al,

2023), increased ruptures of *dLamB-RNAi* nuclei may impact the activation of DNA damage repair pathways within these leukocytes.

Control leukocytes are thus largely able to resist confinement-associated nuclear ruptures and MN formation, but this protection may rely upon lamina adaptation, as leukocyte nuclear defects are significantly increased if leukocytes fail to elevate dLamB upon transition to vessels in vivo. While dLamB-deficient leukocytes experienced the most frequent nuclear herniations on transition to vessel confinement, reduced dLamB also caused more nuclear herniations within leukocytes in the unconstrained 18 h APF wing environment (Fig. 5M). Given that phagosomes within 18 h APF leukocytes often indent the nucleus (Fig. 2), dLamB might afford nuclear protection in vivo from internal phagosomal pressure; this is consistent with recent in vitro work demonstrating a protective role for the nuclear lamina against lipid droplet-induced rupture (Ivanovska et al, 2023).

A-type lamins (including dLamC) could have conflicting roles during confined migration, with potential trade-offs between nuclear deformability and cell survival (Pfeifer et al, 2019; Yamada and Sixt, 2019). This may explain why neutrophils (with minimal A-type lamins) (Olins et al, 2001) can invade almost any tissue but die within days (Pillay et al, 2010). Here, we find leukocyte-specific reduction in dLamC levels increased nuclear abnormalities (e.g. herniations) and MN formation as leukocytes transitioned to veins, albeit not significantly, above controls (Fig. 5M–O; Appendix Fig. S2E). Conversely, leukocytes with elevated dLamC experienced more nuclear abnormalities in vivo during confined migration, specifically within 75 h APF vessels, than those exhibited by control or Lamin-deficient leukocytes (Fig. 5M; Appendix Fig. S2D); this suggests that leukocytes must maintain a precise balance of A-type Lamins as too much or too little can have dramatic effects on nuclear health. Nevertheless, leukocytes appear to partially buffer changes in A-type Lamins in vivo, as we observed a compensatory increase in dLamB in leukocytes with experimentally elevated dLamC; this could thus maintain a more 'control-like' Lamin stoichiometry and counter more severe defects that could otherwise have been caused by dLamC-induced nuclear rigidity.

## Leukocyte extravasation across vessel walls in vivo triggers actomyosin-associated nuclear deformation and micronuclei formation

To perform key roles in immune surveillance and inflammation, mammalian leukocytes in vivo rely on their remarkable ability to squeeze between closely apposed cells, such as those lining the blood or lymphatic vessel walls (Nourshargh and Alon, 2014). *Drosophila* leukocytes can also extravasate across the walls of the narrow 75 h APF wing vessels during immune surveillance (Fig. 6A,B) and in response to exogenous tissue damage (Thuma et al, 2018). Strikingly, we find that leukocyte 'extravasation' between tightly-apposed cells making up the vessel walls in vivo triggered dynamic nuclear deformation, as observed using nuclear RFP (Fig. 6C,D), RFP-tagged Histone2A (Fig. 6E and Movie EV9) and RFP labelling of endogenous dLamB (Fig. 6F; Appendix Fig. S3A). Using our computational pipeline, we quantified nuclear deformation of leukocytes migrating within, and transmigrating across, 75 h APF wing vessels (Fig. 6G–I). Transmigrating leukocytes endured significantly greater nuclear deformation than leukocytes migrating within the vessel lumen (Fig. 6G) and reached much smaller nuclear diameters (as low as 1.5 μm, Fig. 6H)

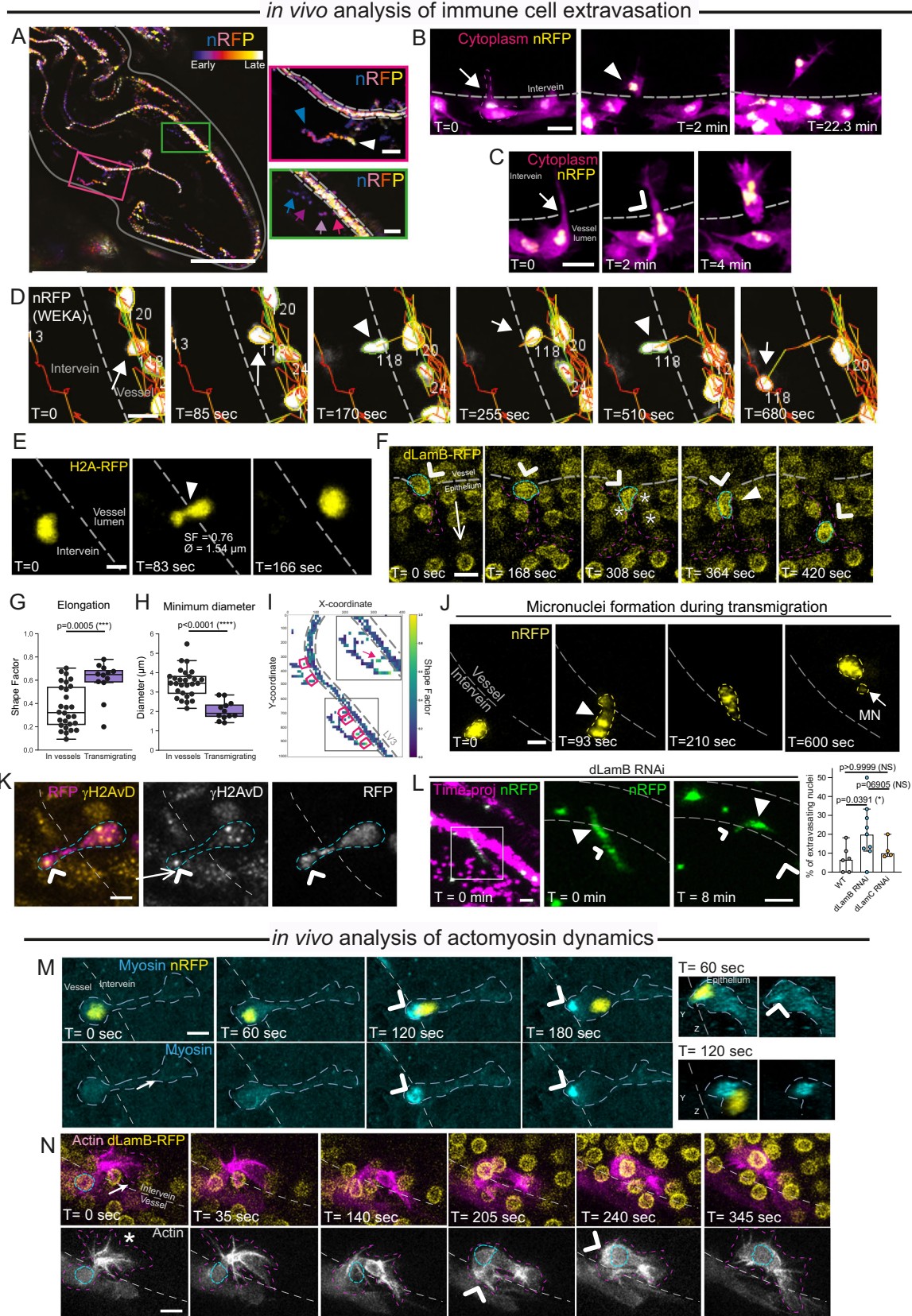

**Figure 6. Exaggerated nuclear deformation and micronuclei formation in vivo during leukocyte transmigration across vessel walls.**

(A) Temporally colour-coded z-projection of hemocyte (nuclear) migration tracks over 5 h (72–77 h APF), with wing outlined (grey). Red inset indicates hemocyte track that starts (blue arrowhead) and ends (white arrowhead) outside the vessel. Green inset indicates a track that enters a vessel (arrows). (B) Time-course of hemocyte probing the vessel wall (arrow) before extravasating (arrowhead) into the intervein space. (C) Hemocyte extravasating from a vessel (arrow), with nuclear squeezing (arrowhead). (D) Hemocyte nucleus (arrow, WEKA probability map in white) extravasating (arrowhead) from 75 h APF vessel. Nuclear tracks in multicolour, numbers refer to tracking ID. (E, F) Hemocyte nuclear chromatin (E, yellow) and nuclear lamina (F, yellow) squeezing (arrowhead) during transmigration. (G, H) The nuclear Shape Factor (G) and minimum nuclear diameter (H) of 75 h APF hemocytes within vessels or during transmigration, as measured from a single 'snapshot'. Boxplots with median and interquartile range (IQR, 25th to 75th percentile), with whiskers extending to the furthest data point within 1.5 times the IQR, individual values as black dots. $N = 3$ and 4 wings, and $N = 28$ and 12 nuclei for hemocytes in vessels or during transmigration. Mann-Whitney U test was used to calculate significance. (I) Heatmaps of average SF value within each square over 2 h of tracking (white = no measured value); rectangles highlight regions where nuclei have extravasated. (J) MN formation (arrowhead) during transmigration across vessel wall. (K) DNA damage (arrow, γH2AvD punctae, yellow) in nucleus of extravasating hemocyte (magenta, RFP). (L) Nuclear ruptures (arrows indicate cytoplasmic nRFP; arrowheads indicate nuclear nRFP) increase during extravasation following dLamB-RNAi; bar chart shows median and 95% CI ($N = 6$, $N = 9$ and 4 wings for control, dLamB-RNAi and dLamC-RNAi, respectively); Kruskal–Wallis with multiple comparisons test used to calculate significance. (M) Rear-localised Myosin II during hemocyte (nucleus yellow, Myosin II cyan) extravasation; sideview of nucleus on right. Arrowheads label Myosin II around nucleus. Hemocyte is outlined in cyan, vessel wall in grey. (N) Leading-edge protrusions (asterisk) in an extravasating hemocyte (Actin in magenta and grey, dLamB in yellow), followed by rear contractions (arrowhead). Nucleus is outlined in cyan, vessel wall in grey. Grey dashed line marks approximate vessel wall location. Long arrows indicate the direction of migration. Genotypes used: Srp > nRFP; Crq > GFP (A–C), Srp > nRFP (D, G–J, K), Srp > GFP, SrpH2AmChe (E), dLamB-tagRFP; SrpGMA (F, N), Srp > nRFP;UAS-dLamB-RNAi (L), Srp > nRFP; Sqh^{AX3}; sqh-Sqh-GFP (M). Data information: Scale bars represent 100 µm (A), 30 µm (A, inset), 10 µm (B, C), 2.5 µm (D, E), 5 µm (F, J–N). Source data are available online for this figure.

suggesting that leukocytes faced even tighter physical constraints as they squeezed across vessel walls; this differential spatial pattern of nuclear deformation could be visualised across the 75 h APF wing using a heatmap displaying mean nuclear Shape Factor (Fig. 6I). Such extreme nuclear deformation during transmigration was also associated with MN formation (observed as nuclear RFP budding from the nucleus, Fig. 6J) and DNA damage (observed via γH2AvD immunostaining, Fig. 6K). While extravasation was rarely associated with nuclear ruptures in control leukocytes, the loss of the B-type nuclear Lamin caused a significant increase in extravasation-associated ruptures (Fig. 6L; Appendix Fig. S3D).

Nuclear deformation during leukocyte extravasation was associated with a rear-localised pulse of Myosin II (Fig. 6M; Appendix Fig. S3B and Movie EV10), similar to that observed during leukocyte patrolling of the vessel lumen (Fig. 2); strikingly, Myosin II accumulated around the rear of the nucleus, with the greatest intensity during nuclear deformation (arrowhead, Fig. 6M). Live-imaging of Actin dynamics revealed that Actin accumulated at the rear of the nucleus during transmigration, although filopodial-like actin protrusions also formed at the leading edge, suggesting that leukocytes might use leading edge protrusions to 'pull' themselves across the vessel wall (Fig. 6N; Appendix Fig. S3C), an event we previously demonstrated to be integrin-dependent (Thuma et al, 2018).

## Leukocyte dynamics during physiological confinement in vivo are influenced by the malleability of adjacent tissues

To date, in vitro studies exploring nuclear dynamics in confined cell migration have focused on molecular mechanisms governing nuclear plasticity intrinsic to the migrating cells (Kameritsch and Renkawitz, 2020; Yamada and Sixt, 2019). However, cells migrating within physiological environments in vivo are frequently constrained by interactions with nearby cells or tissues. In these in vivo scenarios, the different biophysical properties of the interacting cells might define which cells will ultimately adapt (e.g. deform) to enable passage; such decision-making will undoubtedly involve complex reciprocal biomechanical interactions between adjacent cells that are currently challenging to accurately replicate in vitro.

Here, we demonstrate that confined leukocyte migration in vivo involves non-autonomous interactions between multiple cells or cell types (Fig. 7). Within 40 h APF wings, leukocytes routinely navigate passed each other within the vessel-like channels (Fig. 7A and Fig. 1H). Strikingly, we find that nuclear deformation often occurred within one, but not both, of the passing leukocytes (Fig. 7A and Movie EV11). Nuclear deformation in these passing leukocytes was associated with Myosin II accumulation around the nuclear periphery (Fig. 7A), specifically within the leukocyte exhibiting nuclear deformation, suggesting that myosin-associated nuclear deformation might enable cell-cell passing during confinement in vivo. Similarly, Actin was enriched around the squeezing leukocyte nucleus as it passed the large nerve within the wing vessel lumen (Fig. 7B), similar to that observed for Myosin II during leukocyte extravasation from vessels. The ability of leukocytes in vivo to successfully navigate passed one another in confining environments likely necessitates a finely-tuned nuclear lamina to support nuclear plasticity and resilience; indeed, we find leukocytes deficient in dLamB develop deformation-associated nuclear defects (including MN formation, Movie EV8) during passing events within the vessel lumen.

Leukocyte migration within the narrow, convoluted vessels of 75 h APF wings was also frequently associated with deformation of the cells making up the channel wall (Fig. 7C) resulting in channel dilation (Fig. 7D,E and Movie EV12). Strikingly, smaller channels underwent proportionally larger deformations during leukocyte nuclear passage (Fig. 7F). Moreover, the channel diameter during nuclear transit closely approximated the diameter of the leukocyte nucleus, suggesting that channels may deform precisely to accommodate the size of the nucleus (Fig. 7G,H). Observations of vessel and leukocyte dynamics from time-lapse imaging may even suggest that the extent of channel wall deformation might inversely correlate with that of the leukocyte nucleus (Fig. 7D), with deformation of the wall occurring when the nucleus of the passing leukocyte was particularly large and relatively round. Overall, our data suggest that leukocyte deformation in vivo is thus a complex interplay between how much an individual leukocyte can adapt its morphology (e.g. squeeze its nucleus) and the resistance provided by cells within the surrounding microenvironment. Similar

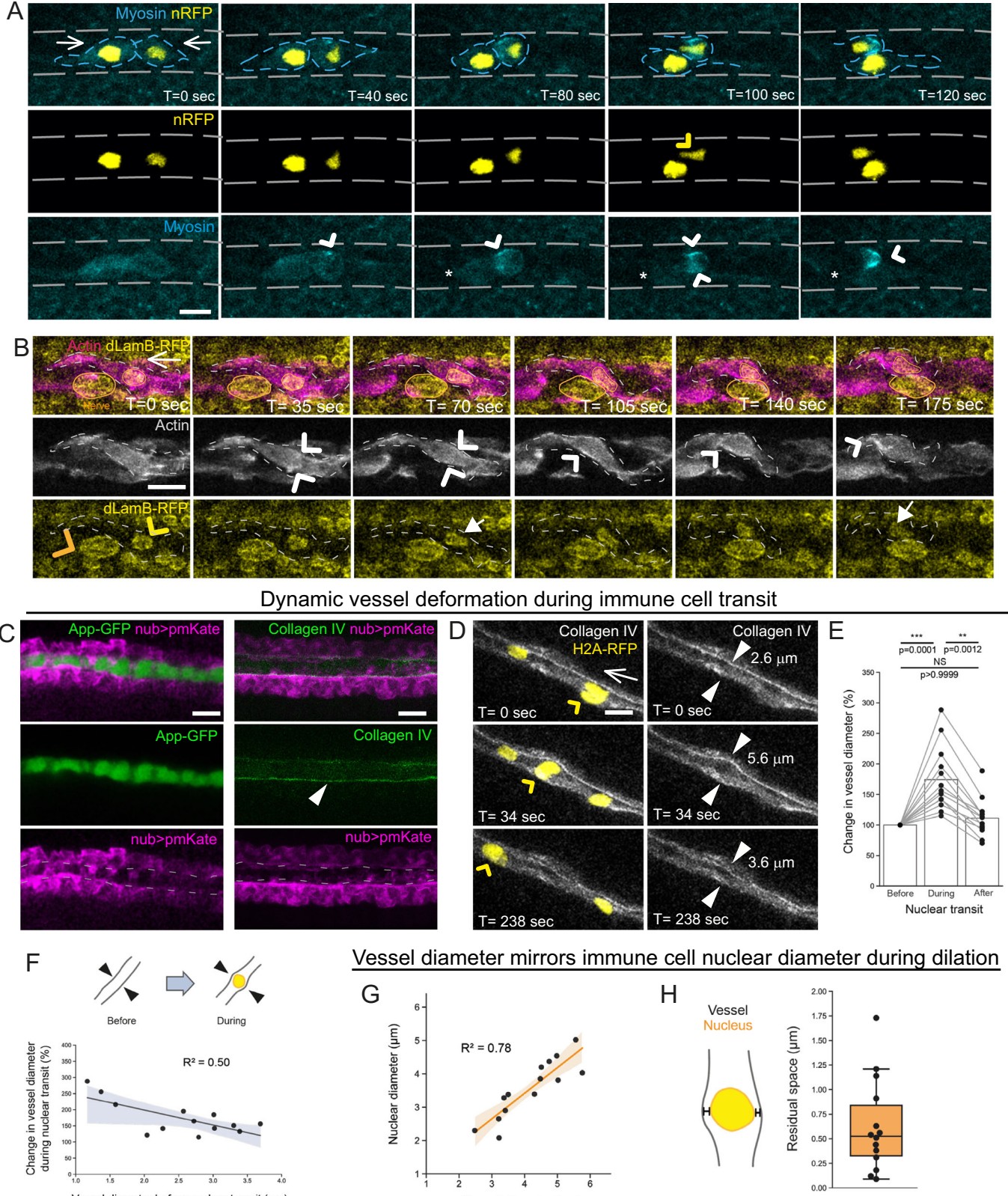

Dynamic vessel deformation during immune cell transit

Vessel diameter mirrors immune cell nuclear diameter during dilation

**Figure 7. Non-autonomous interactions of leukocytes with surrounding cells and tissues.**

(A) Two hemocytes passing each other in a 40 h APF wing vessel (nucleus in yellow, Myosin II in cyan). White arrowheads label perinuclear Myosin II accumulation, yellow arrowheads indicate squeezing nucleus, asterisk labels lack of Myosin II at the cell rear. Approximate vessel locations are marked with a grey dashed line, with hemocyte outlines in dashed cyan. (B) Hemocyte (dLamB in yellow, actin in magenta and grey) navigating passed a nerve cell (nerve nucleus outlined in orange, orange arrowhead) in a 75 h APF vessel. Hemocyte nucleus (yellow arrowhead) squeezes (white arrow) with dynamic actin accumulating around and behind the nucleus. Arrows indicate direction of migration, hemocyte outlined in grey. (C) 75 h APF vessel structure, with vessel wall cells (magenta, labelled with palmitoylated-mKate) separating the extracellular space (green, App-GFP) and lined by extracellular matrix (arrowhead, green; Vkg-GFP). (D, E) Hemocytes migrating in a 75 h APF vessel (Collagen IV in grey, hemocyte nucleus in yellow). Arrowhead points to the nucleus of interest. Arrowheads label the dynamic changes in vessel diameter as the hemocyte passes through; note that vessel widths were measured from the z-plane in focus with the hemocyte nucleus. Quantification depicts the change in vessel diameter during deformation events, relative to pre-hemocyte transit diameter, with individual vessel diameters shown with dotted lines and means shown with bars. (F) Illustration and quantitative correlation of vessel diameter before hemocyte nuclear transit with the relative change in vessel diameter during nuclear transit. (G) Correlation of vessel and nuclear diameters during nuclear transit events resulting in vessel dilation. For (F, G), individual measurements (black dots), linear regression, 95% confidence interval and coefficient of determination ($R^2$) are shown. Pearson correlation coefficient, $r = -0.71$, $p = 0.0066$ (**) (F) and $r = 0.8804$, $p < 0.0001$ (****) (G). (H) Illustration and boxplot depicting the residual space during hemocyte transit, measured by deducting nuclear diameter from vessel diameter. Boxplot with median and interquartile range (IQR, 25th to 75th percentile), with whiskers extending to the furthest data point within 1.5 times the IQR, individual values are depicted with black dots. Genotypes used: nRFP/Sqh$^{Ax3}$;sqh-Sqh-GFP (A), dLamB-tagRFP;SrpGMA (B), nub>palm-mKate; App-GFP (C), nub>palm-mKate; Vkg-GFP (C), SrpH2AmChe /Vkg-GFP (D–H). Data information: Scale bars represent 5 μm (A, C, D) or 10 μm (B). N = 13 (E, F) or 14 (G, H) transit events from 6 wings (E–H). One-way ANOVA (Friedman) with Dunn's multiple comparisons test (E) or Pearsons's correlation (F, G) was used to calculate significance. Source data are available online for this figure.

phenomena may occur in mammalian tissues, where endothelial deformation can occur downstream of the intravascular pressure (Manley et al, 2020).

## Discussion

Despite recent advances in our understanding of how nuclear mechanobiology governs the migration of cells (particularly cancer cells and leukocytes) through confining microenvironments, our current knowledge is derived from in vitro studies of cells migrating through micron-scale fabricated (artificial) constraints or 3D matrices (Raab et al, 2016; Thiam et al, 2016; Wolf et al, 2013; Renkawitz et al, 2019). Detailed mechanistic analysis of native cells migrating under physiological confinement within their in vivo tissue context has been challenging due to experimental challenges associated with traditional mammalian models. However, invertebrate models are now emerging as powerful new tools to dissect the molecular cell biology underlying developmentally programmed confined cell migrations in vivo (Bone et al, 2016; Penfield and Montell, 2023; Belyaeva et al, 2022). Here, we demonstrate the *Drosophila* wing offers new opportunities to probe how mature leukocytes (particularly their nuclei) dynamically adapt as they move within increasingly confined microenvironments in vivo during immune surveillance.

By integrating live time-lapse imaging of leukocyte behaviour in vivo with cell-type specific genetic perturbation within intact living *Drosophila*, we reveal that leukocytes undergo increased nuclear deformation (both frequency and amplitude) as they become confined within narrow vessel-like channels; this deformation is accompanied by elevated DNA damage and further exaggerated as leukocytes squeeze (extravasate) out across vessel walls. Such leukocyte nuclear plasticity is supported by a precisely balanced lamina composition (with relatively low dLamC compared to dLamB) which changes dynamically as leukocytes experience more confined environments in vivo (with elevated dLamB for increased deformability). Leukocytes in vivo are particularly adept at optimising their nuclear lamina, as experimental elevation of dLamC triggered compensatory upregulation of dLamB. Crucially, this nuclear plasticity in turn supports effective

immune surveillance in vivo, as long-term live-imaging reveals leukocytes that fail to soften their nuclear lamina traffic over significantly smaller distances during confinement.

Leukocyte lamina adaptation is not only required to enhance nuclear plasticity upon confinement, but provide protection from more severe nuclear defects, such as nuclear envelope rupture or MN formation. Indeed, experimental reduction of leukocyte dLamB (to pre-vessel levels) led to significantly more nuclear ruptures and MN during migration within the vessel-like channels. Until now, MN formation has largely been observed during cell division and migration-associated MN formation during interphase has been restricted to cancer cells (Denais et al, 2016; Irianto et al, 2017; Krupina et al, 2021). Our novel observation of MN formation live in vivo within leukocytes undergoing nuclear deformation during immune surveillance and transmigration across vessel walls, opens up *Drosophila* to explore how physiological MN formation affects cellular health and long-term survival. Since chromatin escape following MN rupture can trigger inflammatory signalling (e.g. via cGAS-STING and interferon production) even in non-immune cells (MacKenzie et al, 2017; Kirsch-Volders et al, 2020; Gekara, 2017), we envision MN formation within leukocytes may affect their inflammatory state (and long-term behaviour) and promote systemic inflammation.

While in vitro studies have revealed key molecular mechanisms governing fast-acting nuclear biomechanics, the long-term effects of physiological confinement (downstream of nuclear deformation) on cellular health, function and survival remains more elusive. Our new *Drosophila* model could offer unique opportunities to explore these phenomena. The nuclear lamina, as well as providing mechanical stability, plays roles coordinating several nuclear processes (including DNA replication, DNA repair and transcription) by regulating the 3D organisation of chromatin (Kim et al, 2019; Etourneaud et al, 2021; Reilly et al, 2022); thus physiological adaptation of the nuclear lamina likely has dramatic downstream effects that may alter leukocyte phenotype and function. Chromatin itself may play key roles in cellular adaptation to mechanical cues (Nava et al, 2020), as recent in vitro work suggests chemokine signals provide migratory advantages during 3D migration through complex environments by modulating chromatin compaction (Calì et al, 2022; Wang et al, 2018). Intriguingly, epithelial cell nuclei can

rapidly soften in response to applied force, which may offer mechanoprotection (Nava et al, 2020).

Although there has traditionally been considered a trade-off between nuclear deformability and cell survival (as evidenced by the extremely short lifetime of neutrophils (Pillay et al, 2010)), our work suggests highly migratory cells may, in fact, utilise specific molecular adaptations to increase their stress resilience and enable prolonged survival on their tortuous journeys. Moreover, since chromatin accessibility can be affected as cells move through constraints in vitro (Jacobson et al, 2018; Hsia et al, 2022; Fanfone et al, 2022), confined cell migration in vivo and the dynamic lamina adaptations associated with it could, in turn, have dramatic long-term effects on gene transcription. In this way, applied mechanical force could further prime or reprogramme leukocytes to optimise their motility in restricted microenvironments in vivo.

A key advantage of our new *Drosophila* model is that it enables the concurrent study of multiple physiological factors on nuclear biomechanics, both those intrinsic to the migrating cell (e.g. lamina composition and large organelles e.g. phagosomes) and those in the extracellular environment. Recent in vitro work has shown how lipid droplets can deform the nucleus (Ivanovska et al, 2023), suggesting that highly phagocytic cells (such as leukocytes) may need to balance confined migration with the increased internal pressure created by phagocytosis to prevent excessive nuclear deformation. The *Drosophila* wing model also enables the study of cellular dynamics in response to varying environments with different degrees of confinement, including moderate confinement provided by the narrow, convoluted wing vessels and more severe confinement during transmigration across vessel walls. While studies of confined cell migration in vitro demonstrate nuclear deformation (and even rupture) during migration through micron-scale constraints, our study reveals that cells migrating within physiological settings in vivo undergo nuclear deformation even under more moderate confinement. Crucially, our data suggest nuclear squeezing is still advantageous in these environments as lamina perturbations that reduced nuclear deformation led to less effective immune surveillance. Nevertheless, we find leukocytes that extravasate across vessel walls deform nuclei even more dramatically, opening up our model to study more severe confined cell migration in vivo.

Cells migrating in vivo encounter highly complex and locally heterogenous microenvironments (Weigelin et al, 2012), which not only exhibit ever-changing material properties but frequently contain multiple (often competing) guidance cues (McDonald and Kubes, 2011; Foxman et al, 1997). Our data suggest leukocyte nuclear dynamics in vivo adapt upon transition to migrating within narrow channels, which could reflect both biomechanical changes associated with increased confinement and other environmental changes associated with vessel formation (such as altered chemical signalling or adhesion molecule expression). In fact, emerging work suggests mechanical cues may play important roles in guiding homoeostatic navigation by sensitising leukocytes to chemotactic cues and increasing migratory capacity (Alraies et al, 2024). Dendritic cells in vitro make precise pathfinding decisions when confronted with locally competing guidance cues, which requires Myosin II-mediated nuclear re-orientation to enable coordinated cellular and nuclear movement (Kroll et al, 2023). Our in vivo *Drosophila* model enables the exploration of how multiple cues (e.g. chemical signals, adhesion molecules and

mechanical constraints) are integrated at the molecular level to control cell (and nuclear) behaviour. Migrating cells in vivo also experience direct physical interactions with adjacent cells or tissues, which might in turn shape cell behaviour. Here, a novel observation is that migrating immune cells within particularly narrow vessels also dynamically deform the tissue (e.g. vessel wall) with which they interact, a phenomenon that has previously been challenging to study. We also find that vessel-confined leukocytes regularly squeeze passed other migrating leukocytes (triggering Myosin-II contractility and nuclear deformation), which can even trigger nuclear instability (MN formation) in cells with suboptimal nuclear lamina.

With unparalleled experimental tools unavailable in mammalian systems, we envision this in vivo *Drosophila* model will accelerate our mechanistic understanding of cell migration in complex and confined environments. Our data already suggests the biomechanics of leukocyte migration in vivo are shaped by both the cell biology of the individual leukocyte and the physical properties of the surrounding microenvironment. In the long-term, we envision improved understanding of the mechanisms underpinning cellular and nuclear adaptation to confined migration, and its relevance to both physiological and pathological processes, will not only impact treatment of inflammatory disorders, but also offer insight for anti-metastatic therapies. Indeed, adaptive mechanisms for confined migration (including Lamin A/C silencing) could aid cancer cell motility through small tissue spaces (Irianto et al, 2016).

# Methods

## Reagents and tools table

| Reagent/Resource | Reference or Source | Identifier or Catalog Number |
|---|---|---|
| **Experimental models** | | |
| *Drosophila melanogaster (genotypes listed below):* | | |
| Act5c-gal4 | Bloomington Drosophila Stock Center (NIH P40OD018537) | #3954 |
| btl-Gal4,UAS-Actin-GFP | Bloomington Drosophila Stock Center | #8807 |
| UAS-Rab7-GFP | Bloomington Drosophila Stock Center | #42706 |
| UAS-Shi^TS | Bloomington Drosophila Stock Center | #44222 |
| LamC[G00158] (dLamC-GFP) | Bloomington Drosophila Stock Center | #6837 |
| UAS-lamGFP | Bloomington Drosophila Stock Center | #7378 |
| UAS-RedStinger (UAS-nRFP) | Bloomington Drosophila Stock Center | #8545 |
| *UAS-dLamB-RNAi 1* | Bloomington Drosophila Stock Center | #57501 |
| *UAS-dLamB-RNAi 2* | Vienna Drosophila Resource Center | #107419 |
| Vkg-GFP | Bloomington Drosophila Stock Center | #98343 |
| elavC155-Gal4 | Bloomington Drosophila Stock Center | #458 |

| Reagent/Resource | Reference or Source | Identifier or Catalog Number |
|---|---|---|
| Simu-CytGFP; 10xUAS-IVS-myr::tdTomato | Bloomington Drosophila Stock Center | #32221 |
| sqhAx3;sqh-Sqh-GFP | Bloomington Drosophila Stock Center | #57144 |
| UAS-lamC-RNAi | NIG-FLY (National Institute of Genetics, Japan) | #10119R-1 |
| Srp-3xmCherry | gift from Daria Siekhaus Ref: Gyoergy et al (2018) | |
| Srp-3xH2A-mCherry | gift from Daria Siekhaus Ref: Gyoergy et al (2018) | |
| Srp-Gal4 | gift from Katja Bruckner Ref: Brückner et al (2004) | |
| Crq-Gal4 | Bloomington Drosophila Stock Center | #25041 |
| Apolipoprotein-GFP | Vienna Drosophila Resource Center | #v318255 |
| lamDm0-tagRFP | gift from Bruno Monier Ref: Ambrosini et al (2019) | |
| UAS-kaede | gift from Helen Skaer Ref: Bunt et al (2010). | |
| UAS-lamC | gift from Mirka Uhlirova Ref: Gurudatta et al (2010) | |
| **Antibodies** | | |
| anti-dLamC | DSHB | LC28.26 |
| anti-dLamB | DSHB | ADL67.10 |
| anti-Histone H2A Phospho Ser137 | GeneTex | GTX48733 |
| horse Biotinylated Anti-mouse IgG | Vector labs | BA-2000-1.5 |
| anti-mouse Alexa-488 | Jackson Immunoresearch | 115-545-003 |
| Streptavidin Cy5 | Vector labs | SA-1500-1 |
| **Oligonucleotides and other sequence-based reagents** | | |
| Rpl32 primers | 5'-AGCATACAGGCCCAAGATCG-3' and 5'-TGTTGTCGATACCCTTGGGC-3' | |
| Lamin Dm0 primers | 5'-CGTCTGCTCGATGACACAG-3' and 5'-CGACTCGTACATGCGGACATT-3' | |
| Lamin C primers | 5'-ACAGAATCTGGAGGACACCAA-3' and 5'-TGTCGTAAGCGGCAATTTCG-3' | |
| **Chemicals, Enzymes and other reagents** | | |
| Phalloidin AlexaFluor633 | Invitrogen | A22284 |
| Fluoresbrite Polychromatic Red Microspheres | PolySciences | 19508-2 |
| 30 Gauge ½ inch needle | SHD Medical | BD Microlance 3 |
| 37% Paraformaldehyde | Sigma | 8.18708 |
| Mowiol4–88 with DABCO | Sigma-Aldrich | 81381 |

| Reagent/Resource | Reference or Source | Identifier or Catalog Number |
|---|---|---|
| 15% sucrose | Fisher Scientific | 12663076 |
| Triton X-100 | Sigma | X100 |
| Schneiders Insect Medium | Sigma | S0146 |
| Bovine serum albumin, BSA | Sigma | A2153 |
| Concanavalin A (ConA) | Sigma | C2010 |
| PowerUp Syber Green | Thermo Fisher Scientific | A25742 |
| **Software** | | |
| Adobe Illustrator | Adobe | |
| Volocity | PerkinElmer | |
| ImageJ/Fiji | NIH | |
| IMARIS | Bitplane | |
| Huygens Professional | http://svi.nl | |
| Prism | Graphpad | |
| **Other** | | |
| Leica TCS SP8 AOBS | Leica confocal microscope | |
| IXplore SpinSR system | Olympus | |

## *Drosophila* stocks and husbandry

Fly stocks were maintained on Iberian food according to standard protocols (Greenspan, 1997). Adults used for setting up crosses were maintained at 25 °C with controlled humidity unless otherwise stated. Balancers or marker chromosomes were removed whenever possible from experimental samples. The following *Drosophila* stocks were used: Act5>GMA (B#3954), btl-Gal4,UAS-Act5-GFP (B#8807), UAS-Rab7-GFP (B#42706), UAS-Shi^TS (B#44222), LamC [G00158] (dLamC-GFP, B#6837), UAS-lamGFP (B#7378), UAS-RedStinger (also called *UAS-nRFP*, B#8545), *UAS-dLamB-RNAi 1* (B#57501), *UAS-dLamB-RNAi 2* (VDRC #107419), Collagen IV protein trap Vkg-GFP (B#98343) (Morin et al, 2001), elavC155-Gal4 (B#458) (Kurant et al, 2008), Simu-CytGFP(Lin and Goodman, 1994), 10xUAS-IVS-myr::tdTomato (B#32221), sqhAx3;sqh-Sqh-GFP (B#57144) (Royou et al, 2004), nub-Gal4 (B#86108), UAS-mys-RNAi (B#27735), UAS-palm-mKate (B#88540), Histone2A-RFP (B#23651), ubi-Fucci (B#55098) and yw^67 all from the Bloomington Stock Centre (NIH P40OD018537). We also used: UAS-lamC-RNAi (#10119R-1) from NIG-FLY (National Institute of Genetics, Japan), Srp-3xmCherry and Srp-3xH2A-mCherry (gifts from Daria Siekhaus) (Gyoergy et al, 2018), Srp-Gal4 (gift from Katja Bruckner) (Brückner et al, 2004), Crq-Gal4 (gift from Paul Martin) (Kurant et al, 2008), Apolipoprotein-GFP (VDRC #v318255, gift from Anna Franz), lamDm0-tagRFP (also called *dLamB-tagGFP*, gift from Bruno Monier) (Ambrosini et al, 2019), UAS-kaede(10) (gift from Helen Skaer) (Bunt et al, 2010), srp-GMA (gift from Paul Martin), UAS-lamC (gift from Mirka Uhlirova) (Gurudatta et al, 2010).

## In vivo live-imaging of hemocytes

White prepupae (0 h after puparium formation, APF) were collected 18 h, 40 h or 75 h before experiment start, and raised at

25 °C unless stated otherwise. For experiments requiring pupae to be raised at 18 °C or 32 °C, pupae were picked 36 or 14.4 h earlier. Puparium cases were removed using forceps as described (Weavers et al, 2018) taking care not to puncture the wing, before imaging in a glass-bottomed dish with the Leica TCS SP8 AOBS confocal laser-scanning microscope with high magnification oil objective (20x/0.75 or 63x/1.4 HC PL APO CS2) maintained at 25 °C; moisturised paper was used to ensure constant hydration. Nuclear dynamics (e.g. with dLamB-GFP) were imaged as above using the Olympus IXplore SpinSR system (Yokogawa CSU-W1 SoRa spinning disk, Olympus IX83 inverted microscope and Hamamatsu Fusion BT sCMOS cameras) with a 60x/1.3 Silicone objective. For micro-injections, 10 μl of Fluoresbrite Polychromatic Red Microspheres (1:10 in 1X PBS (Sigma-Aldrich)) were injected with an FemtoJet micromanipulator (Eppendorf) equipped with a microinjection needle (Femtotip II). For photoconversion experiments, Kaede-expressing wing hemocytes were photoconverted at 18 h APF (1 min, 405 nm) and imaged at 18 h and 75 h APF (using 488 nm and 561 nm lasers to follow unconverted Kaede and converted Kaede*, respectively). Hemocytes with only punctate Kaede* (red phagosomal aggregates) were considered unconverted. Wings without photoconversion served as negative controls for baseline Kaede* levels.

## Immunostaining and histology of pupal wings

The pupal upper abdomen was punctured with a 30 Gauge ½ inch needle (BD Microlance 3) and fixed as described before (Thuma et al, 2018), using 20 min fixation steps with 8% and then 4% paraformaldehyde (Sigma), and a further dissection of wings with a surgical blade (Swann Morton) prior to blocking. Immunostaining was performed using primary antibodies (anti-dLamC (LC28.26, DSHB, 1:100) or anti-dLamB (ADL67.10, DSHB, 1:200)) and secondary antibodies (1:200 of horse Biotinylated Anti-mouse IgG (Vector labs) and Streptavidin Cy5 (Vector labs)). Wings were mounted with Mowiol4–88 (Sigma-Aldrich) supplemented with DABCO anti-fading agent (Sigma-Aldrich) for imaging. For histology, pupae were dissected and fixed as above, before transferring into 70% ethanol (Fisher Scientific) at 4 °C overnight and embedded in wax. 10 μm sections were stained with hematoxylin and eosin (H&E) (Corrin, 1981) and imaged with an Olympus DP27 5-megapixel camera attached to an Olympus BX53 microscope (20x/0.5 or 40x/0.75 UPlanFL objective).

## Cryo-sectioning and immunostaining of pupal wing sections

Pupae were fixed as above before immersion in 15% sucrose (Fisher Scientific) for 40 min and 30% sucrose for 50 min. The pupal wings were removed, immersed in Optimal Cutting Temperature (OCT, VWR chemicals) mounting media and flash-frozen in liquid nitrogen. 7 μm sections were cut out in a Bright 5040 cryotome and positioned on microscope slides (Superfrost Plus Adhesion Slides, Epredia). Cryosections were then stained as follows: 5 min washes in PBS-TX (1X PBS, 0.1% Triton X-100, both Sigma) before blocking with 1% bovine serum albumin (BSA in PBS) for 60 min and overnight incubation with primary antibody (e.g. 1:500, anti-Histone H2A Phospho Ser137, GeneTex). Primary antibodies were omitted from negative controls. Sections were rinsed with PBS-TX

before incubation with the Secondary antibody (1:200, anti-mouse Alexa-488, Jackson Immunoresearch) or with DAPI (1:1000, Invitrogen). Sections were mounted as above for imaging. For DNA damage analysis, positive control pupae were irradiated with γ-rays from a Caesium137 source for 2 h with an approximate dose rate of 0.42 Gy/minute.

## Immunostaining of primary hemocytes ex vivo

Pupae were collected and dissected as above. Glass-bottomed dishes were coated with 0.5 mg/ml concanavalin A (ConA, Sigma) before rinsing with PBS immediately before use. Fresh supplemented Schneiders (SSM, Schneiders Insect Medium (Sigma) with 10% bovine serum albumin, BSA) was added to the ConA-coated dishes. Pupal wings were removed using forceps and a surgical blade and then submerged in the SSM, before shredding the wings to free hemocytes. After 20 min excess SSM was removed and hemocytes fixed (4% paraformaldehyde for 15 min). Hemocytes were immunostained using primary and secondary antibodies (as described above) as well as Phalloidin AlexaFluor633 (1:200, Invitrogen) before being transferred to PBS for imaging (Karling and Weavers, 2024).

## Flow cytometry-based isolation of hemocytes

18 h APF pupae were submerged in fresh SSM in a watch glass placed on ice and bled with forceps. The carcass was removed, and the cell-SSM mixture was gently homogenised before centrifuging for 10 min at $100 \times g$ at 4 °C. The pellet was resuspended into ice-cold SSM and filtered through a 40 μm cell strainer (Corning) before proceeding immediately with Fluorescence-activated Cell Sorting (BD FACS Aria II, Becton Dickinson). Non-fluorescent extracts were used to create gates to detect nRFP (Red Stinger) positive cells. Hemocytes were sorted into lysis buffer (Qiagen RNeasy Plus Micro Kit) before RNA extraction (Qiagen RNeasy Plus Micro Kit). RNA was precipitated (3 M Sodium Acetate and ice-cold ethanol) before centrifuging ($10,000 \times g$ for 30 min at 4 °C) and resuspending in RNAse-free water. RNA quality was assessed with Tape Station High Sensitivity RNA Screen Tape (Agilent). Equal amounts of RNA were reverse-transcribed (Maxima First Strand Synthesis kit, Thermo Fisher Scientific) and RT-qPCR was performed with the QuantStudio 3 Real-Time PCR System by using PowerUp Syber Green master mix (Thermo Fisher Scientific). The following primers were used: Rpl32: 5′-AGCATACAGGCCCAAGATCG-3′ and 5′-TGTTGTCGA TACCCTTGGGC-3′, Lamin Dm0 (dLamB, lamDm0 primer pair PP18657 (Zirin et al, 2020)): 5′-CGTCTGCTCGATGACACAG-3′ and 5′-CGACTCGTACATGCGGACATT-3′, Lamin C (dLamC): 5′-ACAGAATCTGGAGGACACCAA-3′ and 5′-TGTCGTAAGCGG CAATTTCG-3′. Primer efficiency was measured by following standard protocols from a standard curve. Gene expression was normalised to the housekeeping gene Rpl32.

## General image analysis, visualisation and statistical analysis

Figures were compiled using Adobe Illustrator (Adobe). ImageJ/Fiji (NIH), Volocity (PerkinElmer) and IMARIS (Bitplane) were used to visualise, process, and analyse images (Schindelin et al, 2012). Representative images enhanced to improve visualisation in figures were treated equally to controls. Data-analysis was always

performed using unmodified raw-data and, where possible, blinded prior to analysis. Deconvolution of UAS-dLamB-GFP signal was performed with Huygens Professional (version 7.6.5, http://svi.nl). Statistical analyses were performed using GraphPad PRISM and selected according to the distribution of the data (parametric or nonparametric). PRISM or Python-based packages (NumPy and Seaborn) were used to plot graphs. For normally-distributed data, means and standard deviations are plotted, but for non-parametric data, medians and 95% confidence intervals are plotted.

## Quantification of wing vessel diameters and hemocyte numbers within vessels

Confocal imaging data of Vkg-GFP pupal wings were used to measure vessel widths (the mean of 4–8 diameter measurements per vessel at 40 h APF, or from of a vessel section at 75 h APF) or vessel lengths. The number of hemocytes across a vessel cross-section was quantified for each section of a vessel containing at least one hemocyte. Multiple hemocytes were counted as occupying the same cross-section if the hemocyte cell bodies were aligned. For vessel measurements in Fig. 7, in-focus z-planes of Vkg-GFP labelled vessels were used to measure vessel diameter before, during and after hemocyte nuclear occupation (the latter determined by an increase in nRFP fluorescence as measured from an intensity profile).

## Manual quantification of hemocyte morphology (including phagosomes and Myosin II)

Hemocyte shape descriptors (maximum and minimum cell diameter, cell area, maximum and minimum nuclear diameter) were measured manually with ImageJ using maximum projections of cytoplasmic GFP or nuclear RFP from single timepoints, which was aided by referencing to the individual z-slices for cells in close proximity to other leukocytes. The number and size of phagosomes within hemocytes were quantified from z-slices within a single timepoint with ImageJ/Fiji. For UAS-Shibire$^{TS}$ measurements, H2A-mCherry signal was used to manually outline nuclei from maximum projections to measure nuclear Shape Factor. Hemocyte Myosin II pulses were identified from maximum-projections using intensity profiles across the nucleus to the hemocyte posterior; Myosin II pulse duration and nuclear distance were quantified from sqh-GFP fluorescence intensity with timepoints immediately before and after the Myosin II pulse used for 'no-pulse' comparison. To quantify nuclear instability events, the number of atypical nuclear shapes, ruptures or micronuclei observed throughout a 20 min imaging period were quantified and normalised to the total number of hemocytes within the wing.

## Semi-automated nuclear tracking with MIA plugin in ImageJ/Fiji

Hemocyte nuclei were segmented, tracked and measured with a custom image-analysis pipeline created with the ModularImageAnalysis (MIA) plugin in ImageJ/Fiji (Cross et al, 2024). In brief, nuclei imaged over a 20 min period were first segmented with a custom designed classifier created within the Trainable Weka Segmentation ImageJ/Fiji plugin (Arganda-Carreras et al, 2017). The nuclear probability map was used to further segment the 3D nuclei in MIA, and the nuclei were tracked with the TrackMate-plugin (Ershov et al, 2022). Tracks with fewer than 4 timepoints or outside vessels were excluded from analysis. The segmented nuclei projections (2D) were used to measure nuclear area, Shape Factor (nuclear elongation by using the long and the short axis from the integral over the nuclear area) (Olenik et al, 2023) and measurements related to nuclear migration. Hemocyte migration track length and Euclidean distance measurements were standardised to 15 min. Circularity of representative nuclear shapes were measured with Volocity and track mean nuclear circularity with MIA plugin in ImageJ. To create heatmaps of hemocyte tracks, the centroid X and Y coordinate of the nuclear tracks created with the MIA plugin were used to identify the Shape Factor of each nucleus that passes a 17 pixel × 17 pixel (approx. 5 μm × 5 μm) square. The average Shape Factor that passes each square was calculated and projected as a heatmap.

## Quantification of immunostaining data

For in vivo immunostaining analysis, wings were imaged with 1 μm z-slice intervals and hemocyte nuclei identified using the H2A-mCherry signal. Nuclear lamina intensities were measured using the ImageJ line tool (three z-slices and averaged); the background average (mean of 3 line measurements from unstained area at same z-plane) was subtracted from the hemocyte average signal to derive final hemocyte intensity. The same process was repeated for nearby epithelial cell nuclei. To compare hemocytes and epithelium, each hemocyte absolute fluorescence value was normalised to the epithelial average of that age (18 h, 40 h or 75 h APF). For analysis of ex vivo hemocytes, hemocytes were imaged every 0.5μm and lamina fluorescence intensities calculated as above from z-slices for in vivo images (with background fluorescence subtracted from the measured hemocyte intensity). For *RNAi* experiments, the MIA plugin (ImageJ/Fiji) segmented the peripheral nuclear lamina from maximum projections to quantify the average fluorescence intensity, which were normalised to the average control hemocyte fluorescence (processed in tandem). Relative dLamC/dLamB ratios were calculated by first normalising the median dLamC (or dLamB) intensities at 40 h and 75 h APF to the 18 h APF median; normalised dLamC values were then divided by normalised dLamB values of the same age to calculate the relative dLamC/dLamB ratio compared to 18 h APF. To measure γH2AvD levels from cryosections, the MIA plugin (ImageJ/Fiji) segmented hemocyte nuclei in 3D to calculate the average γH2AvD fluorescence intensity within a nucleus. The average negative control fluorescence levels (background) was removed from each individual nuclear fluorescence value within the same batch. Hemocytes that were not fully adherent or were rounded, blebbing or otherwise appeared apoptotic were excluded from the analysis.

# Data availability

No primary datasets have been generated and deposited.

The source data of this paper are collected in the following database record: biostudies:S-SCDT-10_1038-S44319-025-00381-0.

# Peer review information

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

## Acknowledgements

We would like to thank members of the Weavers, Martin and Richardson labs for helpful discussion. We also thank the Wolfson Bioimaging Facility (University of Bristol, UK), Bloomington Stock Centre (University of Indiana, USA) and Vienna Drosophila Resource Centre, for Drosophila stocks. This research was funded by the Wellcome Trust (a Wellcome Trust and Royal Society Sir Henry Dale Fellowship to HW [208762/Z/17/Z]). For the purpose of Open Access, the author has applied a CC BY public copyright license to any Author Accepted Manuscript arising from this submission.

## Author contributions

**Tua Karling**: Data curation; Formal analysis; Validation; Investigation; Visualization; Methodology; Writing—original draft; Project administration; Writing—review and editing. **Helen Weavers**: Conceptualization; Resources; Data curation; Formal analysis; Supervision; Funding acquisition; Investigation; Visualization; Methodology; Writing—original draft; Project administration; Writing—review and editing.

Source data underlying figure panels in this paper may have individual authorship assigned. Where available, figure panel/source data authorship is listed in the following database record: biostudies:S-SCDT-10_1038-S44319-025-00381-0.

## Disclosure and competing interests statement

The authors declare no competing interests.

# Expanded View Figures

**Figure EV1.  Migrating immune cells adapt upon transition into more confined vessel microenvironments in vivo.**

(A) Schematics of wing vessels, labelling lateral veins (LV1-5) and anterior and posterior cross veins (ACV/PCV) at 40 h APF, 75 h APF and adults. (B) Raw data for Fig. 1G; maximum projection of Vkg-GFP pupal wings (grey) at 40 h and 75 h APF. Super-plots depict vessel widths, where solid shapes depict mean vessel diameters and hollow shapes depict individual vessel measurements. (C) Length of LV3 and LV5 vessels at 40 h APF and 75 h APF. (D) Hemocytes per wing at 18 h, 40 h and 75 h APF. (E) Schematic of tracheal and nervous system in 18 h APF wings, with imaging of Collagen IV (grey), trachea (cyan) or nerves (yellow). Wings outlined in grey. (F) Maximum projection of hemocytes (magenta) and trachea (cyan) in a 75 h APF vessel (left), single z-slices illustrate tracheal location above hemocytes (outlined in magenta). (G–J) Hemocyte cytoplasmic roundness (G), aspect ratio (H), minimum diameter (I, cell body) and maximum nuclear diameter (J). Scatter plots show small dots (hemocytes) and large black dots (wing medians), bar chart shows median and 95% CI. (K) Photoconversion of wing hemocytes: UV-exposed Kaede-expressing hemocytes captured with 488 nm (Kaede) and 561 nm (photoconverted, Kaede*). Wing outlined in solid line, abdomen in dashed line. (L) Non-UV exposed hemocytes (Kaede) at 18 h and 75 h APF. Insets depict representative hemocytes. Bar chart shows % of photoconverted hemocytes at 18 h and 75 h APF with median and 95% CI. (M) Hemocyte proliferation rate (quantified as percentage of hemocytes per pupal wing dividing per hour) with median and 95% CI. Genotypes used were *Vkg-GFP* (**B-C**, **E**), *Srp > nRFP* (**D**), *Btl > GFP* (**E**), *Elav>Tom;Simu-GFP* (**E**), *Btl > GFP;Srp-mChe* (**F**), *Srp > nRFP; Crq > GFP* (**G–J**, **M**) and *Srp>Kaede* (**K**, **L**). Data information: Scale bars represent 100 μm (**B**, **L**), 200 μm (**E**), 10 μm (**F**) or 15 μm (**L**, insets). Error bars show mean $+/-$ SD of wings (**B–D**). $N = 3$ wings (**L**) for 18 h and 75 h APF. $N = 4$ and 5 (**B**) and $N = 4$ and 3 (**C**) wings for 40 h and 75 h APF. $N = 9$, 9 and 8 (**D**), $N = 3$, 3 and 5 (**G**, **H**) wings for 18 h, 40 h and 75 h APF, respectively. $N = 3$, 4 and 5 (**I**, **J**) wings for 18 h, 40 h and 75 h APF, respectively. $N = 6$, 6, 6, 6 and 6 (**M**) wings for 18 h, 22 h, 30 h, 40 h and 75 h APF, respectively. $N = 372$ and 169 hemocytes for 18 h and 75 h APF (**L**). $N = 216$, 102 and 136 hemocytes (**G**, **H**) and $N = 93$, 83 and 111 hemocytes (**I**) and $N = 95$, 110 and 98 hemocytes (**J**) for 18 h, 40 h and 75 h APF, respectively. Unpaired t test (**C**), Ordinary one-way ANOVA with Turkey's multiple comparisons test (**E**), One-way ANOVA (Kruskal-Wallis) with Dunn's multiple comparisons test (**G–J** and **M**) or one sample Wilcoxon Signed Rank Test (**L**) was used to calculate significance was used to calculate significance. Source data are available online for this figure.

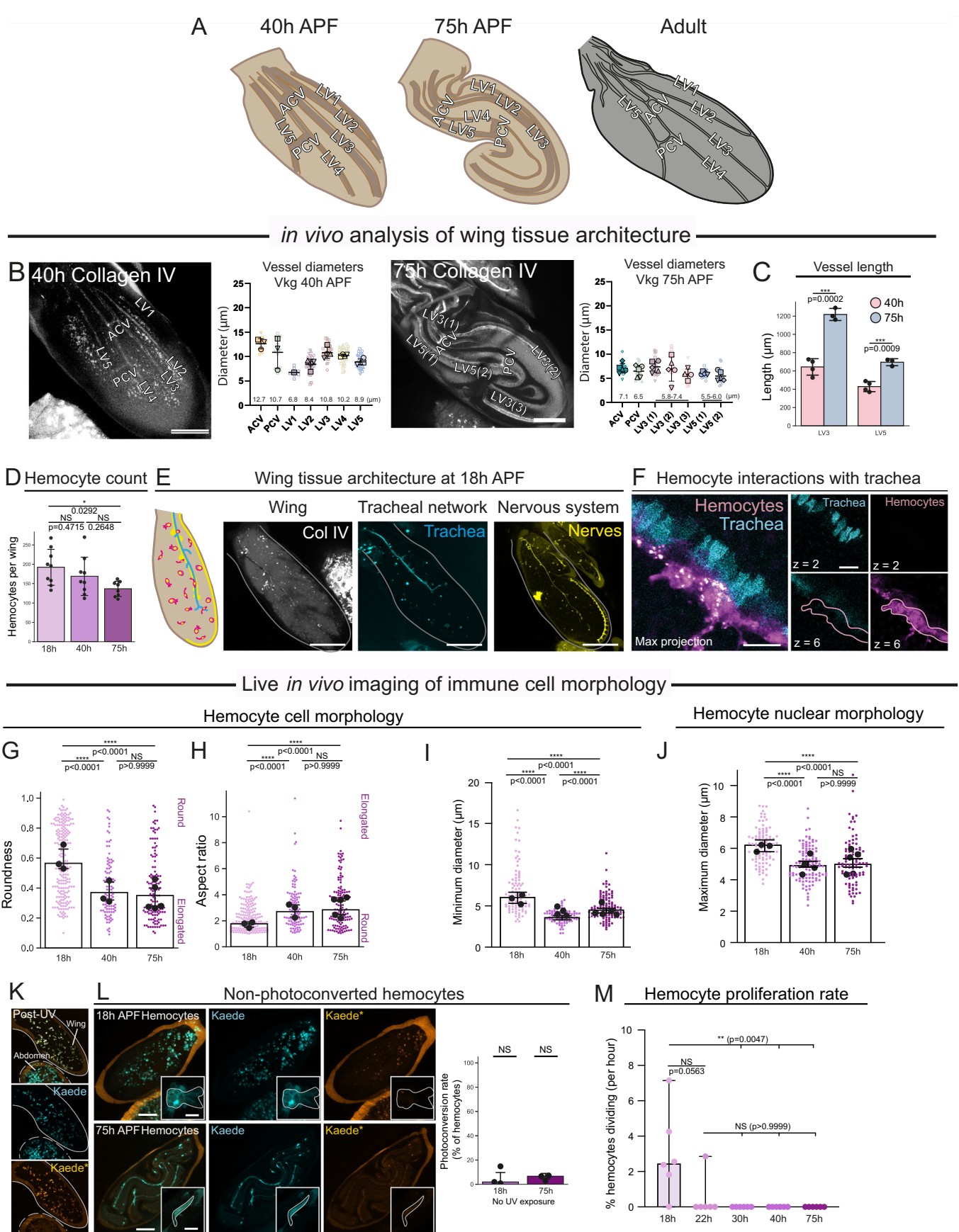

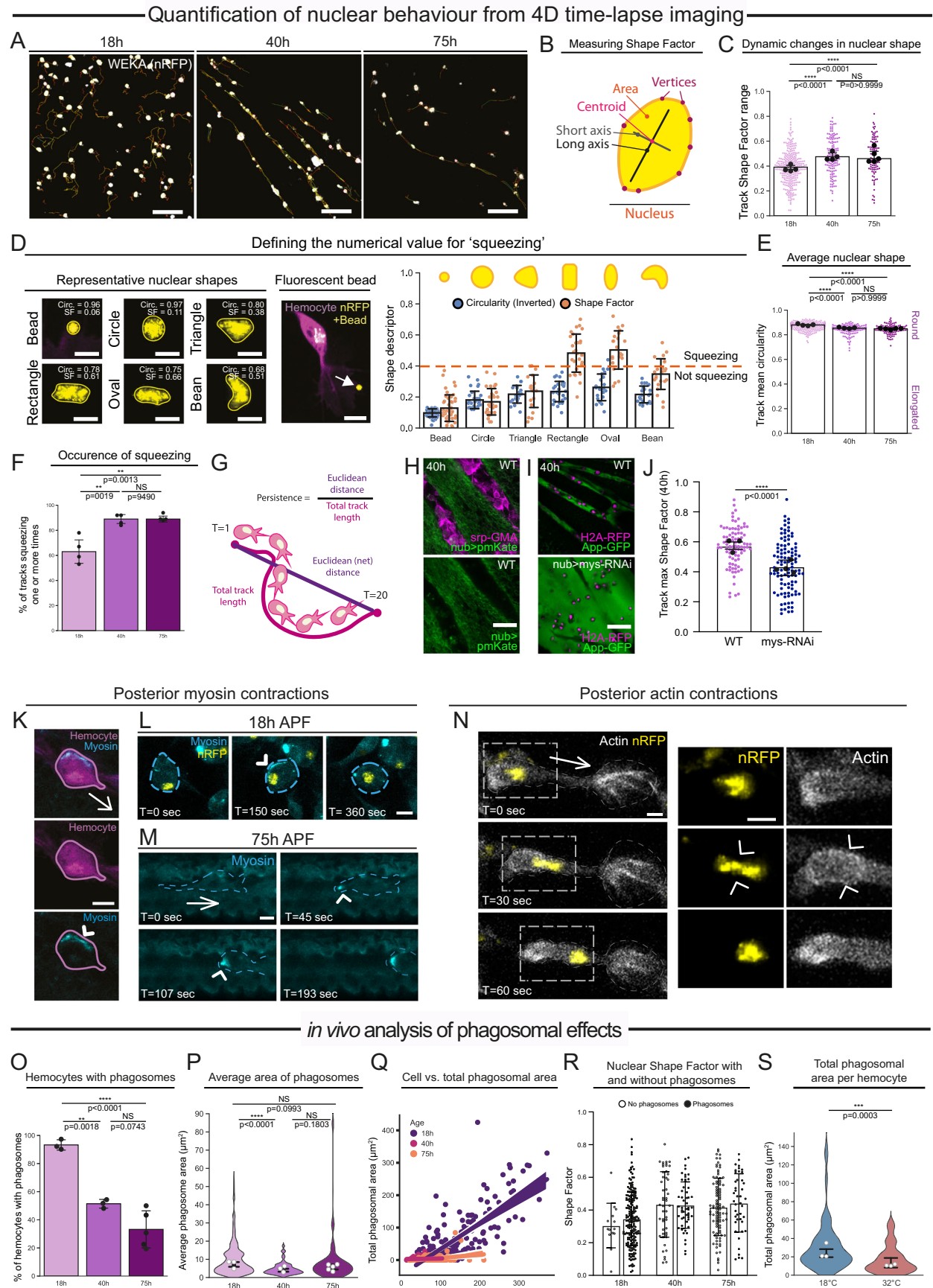

Quantification of nuclear behaviour from 4D time-lapse imaging

**A** 18h 40h 75h WEKA (nRFP)

**B** Measuring Shape Factor

**C** Dynamic changes in nuclear shape

**D** Defining the numerical value for 'squeezing'

Representative nuclear shapes — Fluorescent bead

Bead Circ. 0.96 SF 0.06 / Circle Circ. 0.97 SF 0.11 / Triangle Circ. 0.80 SF 0.38
Rectangle Circ. 0.78 SF 0.61 / Oval Circ. 0.75 SF 0.66 / Bean Circ. 0.68 SF 0.51

Hemocyte nRFP +Bead

**E** Average nuclear shape

**F** Occurence of squeezing

**G** Persistence = Euclidean distance / Total track length

**H** 40h WT srp-GMA nub>pmKate / WT nub>pmKate

**I** 40h WT H2A-RFP App-GFP / nub>mys-RNAi H2A-RFP App-GFP

**J** Track max Shape Factor (40h)

Posterior myosin contractions

**K** Hemocyte Myosin

**L** 18h APF Myosin nRFP T=0 sec / T=150 sec / T= 360 sec

**M** 75h APF Myosin T=0 sec / T=45 sec / T=107 sec / T=193 sec

Posterior actin contractions

**N** Actin nRFP T=0 sec / T=30 sec / T=60 sec — nRFP Actin

*in vivo* analysis of phagosomal effects

**O** Hemocytes with phagosomes

**P** Average area of phagosomes

**Q** Cell vs. total phagosomal area

**R** Nuclear Shape Factor with and without phagosomes

**S** Total phagosomal area per hemocyte

◀

**Figure EV2. Dynamic nuclear shape changes during immune cell migration in vessels in vivo.**

(A) Example hemocyte nuclear migration tracks after processing with the MIA plugin from 18 h, 40 h and 75 h APF. (B) A custom shape measurement 'Shape Factor' used to measure nuclear elongation. Nuclear boundaries inferred from an input shape (WEKA segmented probability map of the nucleus) were used to create a polygon than closely approximates the nuclear shape. The vertices and the integral of the polygon area were used to infer the long and the short axis, then used to calculate a Shape Factor value between 0 (round) and 1 (elongated) (Olenik et al, 2023). (C) The range of Shape Factor values from a 20 min imaging period at 18 h, 40 h and 75 h APF. (D) Representative nuclei for the different shape categories. Image shows a hemocyte (cytoplasm in magenta, nucleus in yellow) and a fluorescent bead (arrow) in a 18 h APF pupal wing. Bar chart shows circularity and Shape Factor (SF) measurements for fluorescent beads and nuclei of different shapes, where small dots are individual nuclei. Dotted line shows the threshold set for squeezing (SF = 0.4); N = > 20 nuclei per category. (E) The average nuclear circularity measured from the same dataset of nuclear tracks as 2D. (F) Percentage of tracks (over 20 min imaging period) with at least one time-point having nuclear SF ≥ 0.4. (G) Schematic showing how persistence, Euclidean (net) distance and total track length are measured. (H–J) Nub-Gal4 drives epithelial-specific gene expression (H); Nub-Gal4 mediated expression of mys-RNAi inhibits vessel formation (I) and reduces hemocyte nuclear deformation in 40 h APF vessels (J, bar chart with median and 95% CI). (K) 40 h APF hemocyte (cytoplasm in magenta, Myosin II in cyan) with rear localised Myosin II (arrowhead); arrow indicates the direction of migration. (L) 18 h APF hemocyte with a Myosin II flash (arrowhead) (nucleus in yellow, Myosin II in cyan). (M) 75 h APF hemocyte with rear Myosin II contraction (arrowhead) (Myosin II in cyan); related multi-channel nRFP and Myosin II images shown in Fig. 2K. (N) Actin networks around hemocyte nucleus (white arrowheads) during squeezing (actin in grey, nucleus in yellow); hemocyte outlined in grey. (O) Percentage of hemocytes with at least one phagosome at 18 h, 40 h or 75 h APF. (P) The average phagosome area per hemocyte, in those with phagosomes. (Q) Correlation between cell area and total phagosomal area. Regression fit for each age is plotted with 95% CI. (R) Nuclear Shape Factor of hemocytes with or without phagosomes at 18 h, 40 h or 75 h APF. (S) Total phagosomal area in individual Shibire$^{ts1}$ hemocytes (for those with phagosomes) as measured from individual z-stacks. Cyan dashed lines mark hemocytes, and arrows indicate migration direction. Genotypes used were: *Srp > nRFP* (A, C–F), *Srp > nRFP; Crq > GFP* (D), *Nub>palm-mKate; srp-GMA* (H), *Nub>mys-RNAi;Srp-H2AmCherry, App-GFP* (I, J), *Nub-Gal4;Srp-H2AmCherry, App-GFP* (I, J), *Srp-mCherry/Sqh$^{AX3}$;sqh-Sqh-GFP* (K), *Srp > nRFP/ Sqh$^{AX3}$;sqh-Sqh-GFP* (L, M), *Srp > nRFP; Srp-GMA* (N), and *Srp > GFP, SrpH2AmChe /UAS-Shi$^{TS}$* (O–S). Data information: Scale bars represent 50 µm (A), 5 µm (D, K–N), 20 µm (H, I). N = 4, 4 and 5 (C, E), N = 4 (F)), N = 3 and 4 (J), N = 3, 3 and 5 (O–R), N = 3 (S) wings for 18 h, 40 h and 75 h APF, respectively. N = 90 and N = 105 hemocytes (J), N = 286, 139 and 102 tracks (C, E), N = 206, 62 and 55 hemocytes, and 1308, 79 and 77 phagosomes measured (O) and N = 216, 102 and 136 hemocytes (P–R) for 18 h, 40 h and 75 h APF, respectively. N = 95 and 32 hemocytes and N = 618 and 146 phagosomes for 18 °C and 32 °C, respectively (S). Scatter plots show small dots (individual nuclear tracks) and large black dots (wing medians), and bar chart shows median and 95% CI (C, E). Bar chart shows mean +/− SD (D, F, L, O, R). Violin plots depict hemocytes, with wing medians in white dots, median and 95% CI shown in black (P, S). One-way ANOVA (Kruskal-Wallis) with Dunn's multiple comparisons test (C, E, P), Ordinary one-way ANOVA with Turkey's multiple comparisons test (F, O), or Mann-Whitney U test (J, S) was used to calculate significance. Source data are available online for this figure.

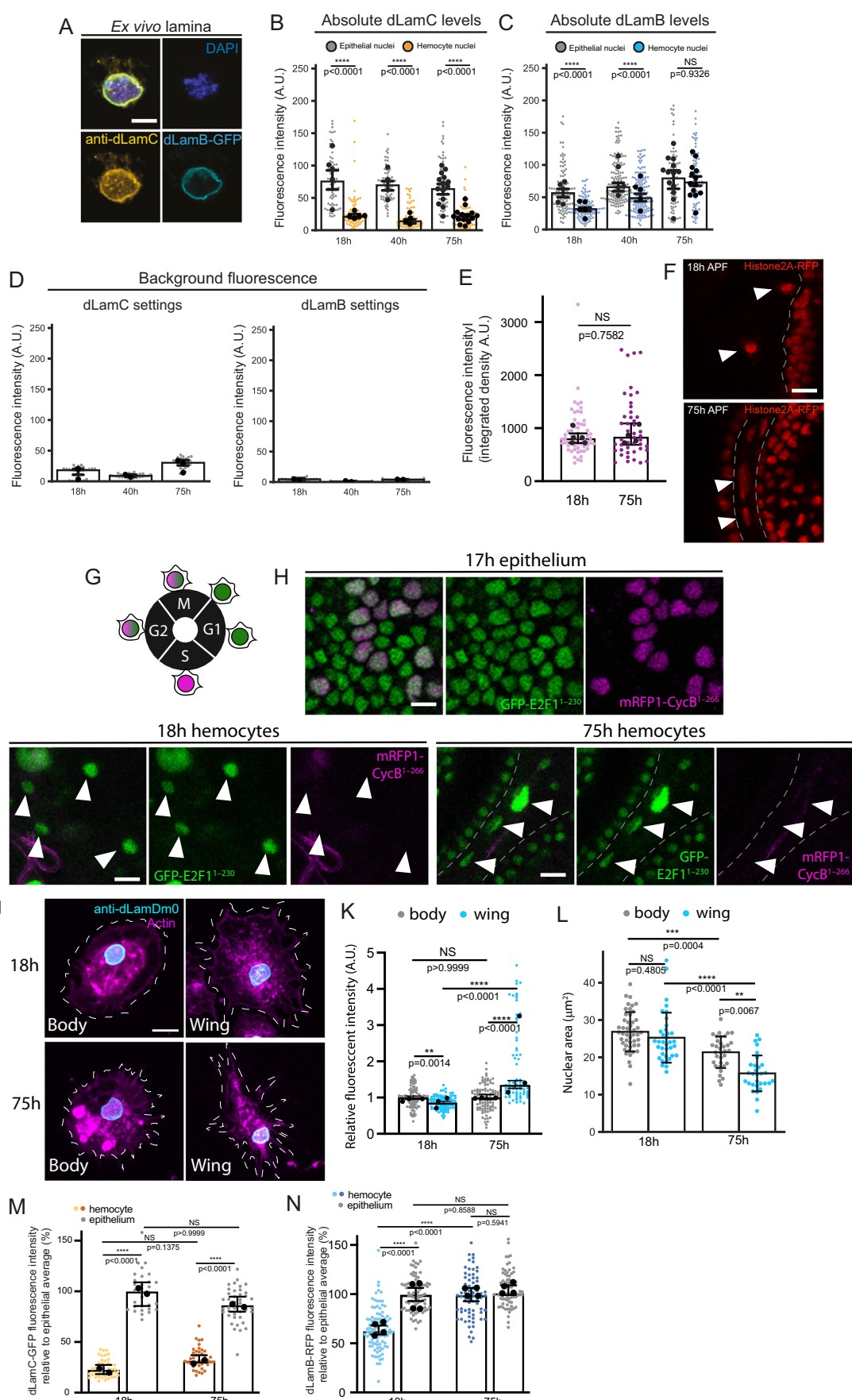

**Figure EV3. Modulation of immune cell nuclear lamina composition upon transition to wing vessels.**

(A) Ex vivo hemocyte nucleus with dLamB-GFP (cyan) stained with DAPI (blue) and anti-dLamC-Alexa488 (yellow). (B, C) Raw anti-dLamC (B) or dLamB (C) fluorescence intensity of hemocyte and epithelial nuclei at 18 h, 40 h and 75 h APF. (D) Background fluorescence levels of hemocytes treated without primary antibody, imaged with anti-dLamC or dLamB settings. (E, F) Raw Histone2A-RFP fluorescence intensity of hemocyte nuclei (arrowheads) at 18 h and 75 h APF. (G–I) Nuclear Fucci labelling of cell cycle stages in epithelium (H) and hemocytes (arrowheads, I). (J–L) Ex vivo anti-dLamB (cyan) and Phalloidin (magenta) staining of wing or body hemocytes (J) with quantification of anti-dLamB fluorescence, displayed relative to that of body hemocytes (K) and hemocyte nuclear area (L). (M, N) Quantification of in vivo endogenous dLamC-GFP (M) or dLamB-RFP (N) fluorescence in hemocytes and adjacent epithelial cells, displayed relative to 18 h APF epithelial average. Genotypes used were: *Srp > nRFP; UAS-dLamB-GFP* (A), and *Srp > GFP, SrpH2AmChe* (B–D and J–L), *H2A-RFP* (E, F), *UAS-Fucci*$^{i[GFP-E2Fl, mRFP1-CycB]}$ (H, I), *dLamC-GFP* (M) and *dLamB-tagRFP* (N). Data information: Scale bars represent 5 μm (A, J) or 15 μm (F, H, I). Scatter plots with small dots (hemocytes) and large black dots (wing section medians), bar chart shows median and 95% CI. $N = 5$, 4 and 14 (B), $N = 5$, 5 and 13 (C), or $N = 2$ wing sections (D) from >8 wings (B, C) or 2 wings (D) (from separate pupae) stained per age (all stains done in one session), and $N = 74$, 67 and 72 nuclei (B), $N = 106$, 151 and 84 nuclei (C), $N = 30$, 27 and 26 nuclei (D, dLamC) or $N = 19$, 33 and 35 nuclei (D, dLamB) measured for 18 h, 40 h and 75 h APF, respectively; $N = 60$ and 46 nuclei (E), $N = 117$ and $N = 103$ body hemocyte and $N = 88$ and $N = 75$ wing hemocyte nuclei (K), $N = 48$ and $N = 34$ body hemocyte and $N = 36$ and $N = 30$ wing hemocyte nuclei (L), $N = 41$ and $N = 40$ hemocyte nuclei and $N = 30$ and 40 epithelial nuclei (M) and $N = 97$ and $N = 76$ hemocyte nuclei and $N = 86$ and 77 epithelial nuclei (N) measured for 18 h and 75 h APF, respectively. Mann-Whitney U test was used to calculate significance for (B–E) and One-way ANOVA (Kruskal-Wallis) with Dunn's multiple comparisons test (K–N). Source data are available online for this figure.

