## [Peer Review File · EMBO Reports]

Immune cells adapt to confined environments in vivo to optimise nuclear plasticity for migration

Helen Weavers and Tua Karling

Corresponding author(s): Helen Weavers (helen.weavers@bristol.ac.uk)

Review Timeline:

Submission Date:	30th Apr 24
Editorial Decision:	14th Jun 24
Revision Received:	22nd Nov 24
Editorial Decision:	17th Dec 24
Revision Received:	5th Jan 25
Accepted:	17th Jan 25

Editor: Achim Breiling

Transaction Report:

Dear Dr. Weavers,

Thank you for the transfer of your manuscript to EMBO reports. I have now received the reports from the three referees that were asked to evaluate your study, which can be found at the end of this email. As you will see, the referees have several comments, concerns, and suggestions, indicating that a major revision of the manuscript is necessary to allow publication of the study in EMBO reports. As the reports are below, and all the concerns need to be addressed, I will not detail them further here.

Given the constructive referee comments, I would like to invite you to revise your manuscript with the understanding that the concerns of the referees must be addressed in the revised manuscript and in a detailed point-by-point response. Acceptance of your manuscript will depend on a positive outcome of a second round of review. It is EMBO reports policy to allow a single round of revision only and acceptance of the manuscript will therefore depend on the completeness of your responses included in the next, final version of the manuscript.

Moreover, please have your revised manuscript carefully edited by a native speaker before re-submission.

- 1) a .docx formatted version of the final manuscript text (including legends for main figures, EV figures and tables), but without the figures included. Figure legends should be compiled at the end of the manuscript text.
- 2) individual production quality figure files as .eps, .tif, .jpg (one file per figure), of main figures and EV figures. Please upload these as separate, individual files upon re-submission.

For papers with not more than 5 main and EV figures, we would publish your manuscript as Report. For a Scientific Report we require that results and discussion sections are combined in a single chapter called "Results & Discussion". Please do this for your manuscript, if the numbers of figures will not increase. For more details please refer to our guide to authors:

<http://www.embopress.org/page/journal/14693178/authorguide#researcharticleguide>

4) a complete author checklist, which you can download from our author guidelines

(<https://www.embopress.org/page/journal/14693178/authorguide>). Please insert page numbers in the checklist to indicate where the requested information can be found in the manuscript. The completed author checklist will also be part of the RPF.

Please also follow our guidelines for the use of living organisms, and the respective reporting guidelines:
<http://www.embopress.org/page/journal/14693178/authorguide#livingorganisms>

5) that primary datasets produced in this study (e.g. RNA-seq, ChIP-seq, structural and array data) are deposited in an appropriate public database. If no primary datasets have been deposited, please also state this in a dedicated section (e.g. 'No primary datasets have been generated and deposited'), see below.

The accession numbers and database should be listed in a formal "Data Availability" section (placed after Materials & Methods) that follows the model below. This is now mandatory (like the COI statement). Please note that the Data Availability Section is restricted to new primary data that are part of this study. This section is mandatory. As indicated above, if no primary datasets have been deposited, please state this in this section

Data availability

8) Regarding data quantification and statistics, please make sure that the number "n" for how many independent experiments were performed, their nature (biological versus technical replicates), the bars and error bars (e.g. SEM, SD) and the test used to calculate p-values is indicated in the respective figure legends (also for EV figures and all those in an Appendix). Please also check that all the p-values are explained in the legend, and that these fit to those shown in the figure. Please provide statistical testing where applicable. Please avoid the phrase 'independent experiment', but clearly state if these were biological or technical replicates. Please also indicate (e.g. with n.s.) if testing was performed, but the differences are not significant. In case n=2, please show the data as separate datapoints without error bars and statistics. See also:
<http://www.embopress.org/page/journal/14693178/authorguide#statisticalanalysis>

9) Please add scale bars of similar style and thickness to microscopic images, using clearly visible black or white bars (depending on the background). Please place these in the lower right corner of the images themselves. Please do not write on or near the bars in the image but define the size in the respective figure legend.

10) Please also note our reference format:

12) We now use CRediT to specify the contributions of each author in the journal submission system. CRediT replaces the

author contribution section. Please use the free text box to provide more detailed descriptions and do NOT provide your final manuscript text file with an author contributions section. See also our guide to authors:
<https://www.embopress.org/page/journal/14693178/authorguide#authorshipguidelines>

13) We would encourage you to use 'Structured Methods', our new Materials and Methods format. According to this format, the Materials and Methods section should include a Reagents and Tools Table (listing key reagents, experimental models, software, and relevant equipment and including their sources and relevant identifiers), uploaded as separate file, followed by a Methods and Protocols section in which we encourage the authors to describe their methods using a step-by-step protocol format with bullet points, to facilitate the adoption of the methodologies across labs. More information on how to adhere to this format as well as downloadable templates (.doc) for the Reagents and Tools Table can be found in our author guidelines (section 'Structured Methods'):

14) Please restrict the keywords to 5 and order the manuscript sections like this, using these names:

Title page - Abstract - Keywords - Introduction - Results - Discussion - Methods - Data availability section - Acknowledgements - Disclosure and Competing Interests Statement - References - Figure legends - Expanded View Figure legends

I look forward to seeing a revised version of your manuscript when it is ready. Please let me know if you have questions or comments regarding the revision.

Yours sincerely,

Referee #1:

The study by Karling and Weavers "Immune cells dynamically adapt to confinement in vivo to enhance nuclear plasticity for Migration" examines a very interesting and emerging topic in the field of cell biology and immunology. They address how changes in the nucleus affect immune cell migration in confined environments. Over recent years, this topic has mainly been studied in microfabricated devices, in particular microchannels with and without constrictions, where mouse or human immune cells moved in a confined space between two stiff surfaces. In these environments, the nucleus as the largest and stiffest organelle was shown to have rate-limiting roles for migration. Here, the authors present a novel, very elegant and much more physiological model system to address questions about the nuclear involvement in immune cell migration. The authors study hemocytes in the *Drosophila* wing system. As presented in this manuscript, this is a beautiful model system, allowing the observation of the same hemocytes over time at different stages of confinement, which result from the developmental stages of the wing vasculature. The authors provide wonderful and convincing imaging of hemocyte migration, including detailed imaging and analysis of nuclear morphology and integrity. Molecularly, the authors investigated expression changes of lamin B and A/C isoforms, showing that fine-tuning of the leukocyte nuclear lamina by balancing A- and B-type lamins aids nuclear deformation and efficient migration. Some interesting findings, e.g. phagosomes can deform nuclei even in non-confined environments or lamin B-deficiency results in decreased DNA damage (Fig. 5P), are presented but remain anecdotal. Instead, the authors show other potential use of this model system for studying nuclear squeezing during the process of extravasation (Fig. 6) or studying the influence of cell-cell deformations in confined environments (Fig. 7).

Overall, I enjoyed reading this manuscript since it presented and discussed the findings of this study very well in the context of current concepts and perspectives of immune cell migration. Figures are executed very well, methods described very well, videos show exemplary cell dynamics of static data represented in the figures. Although not all observations were mechanistically followed up, the paper gives thought-provoking impulses, such as considering the influence of deformability in cells that shape physiological confinements, which cannot be performed very well in existing in vitro systems.

Major comment:

Since the paper is more descriptive than mechanistic, I would have wished that the authors also included some data how depletions of A- and B-type lamins influence the extravasation process, where nuclear squeezing and shape changes are particularly important. Since this has not been addressed in mammalian systems in detail, this study could provide important novel insight into this process from the *Drosophila* hemocyte perspective.

Minor points:

1) Although the paper is really enjoyable to read, it is partly also a bit lengthy. The experimental results are presented in the light of previous research, explaining the rationale of the experiments, which is really nice. However, it feels as if some of these parts were redundant with parts in the introduction and discussion. A bit crisper version of the manuscript would be the presentation of a very good study even better.

2) In Figure 3I: The authors include "stiffer" and "softer" into the figure. Since this is probably based on assumptions from the literature rather than on real measurements, I would suggest to delete it from the figure.

Referee #2:

In vivo imaging of nuclear changes in shape and integrity have been a major challenge to study and relate to lamin levels that possibly change across a key developmental or disease timescale. The hemocytes and epithelial cells studies here provide such an opportunity, and the compelling and convincing studies here add in vivo relevance in the very basic fly model system to various past studies, mostly on isolated cells or tissue. The findings here will be a valuable addition to the literature after addressing a few concerns that temper enthusiasm.

1. Fig.3's analyses of lamin levels indicate the B-type lamin (dLamB) is more variable with time but closer to epithelial levels (in solid tissue) than the A-type lamin (dLamC) that is much lower than epithelial levels. These observations for hemocytes seem similar to those in Shin et al [PNAS 2013: Lamins regulate cell trafficking and lineage maturation of adult human hematopoietic cells] who reported that (i) solid tissue cells (e.g. MSCs) also tend to have much higher lamin-A levels and that (ii) 'B-type lamins vary by ~30-fold whereas lamin-A varies much less'. This relation to human blood lineage cells should be discussed and referenced by the present authors.

2. In human cells, LMNB1 increases 2-fold from G0/G1 to G2/M [Vashisth PNAS 2021], and so measurements of DNA content in the fly system at the later versus earlier timepoints need to be made and considered in relation to the 2-fold increase in dLamB. More DNA in the later nuclei could occur and also explain why there is more DNA damage measured per nucleus; in other words, more DNA provides more substrate for more damage and/or triggers a cell cycle checkpoint [e.g. Pfeiffer MBoC 2016]. Measurements of DNA per nucleus might be made from existing data for DNA or histone intensities.

3. Fig.S5D shows more DNA damage after irradiation only at late times and only for LamB knockdown. Given that these nuclei tend to rupture more per Fig.5N and rupture releases DNA repair factors [Ivanovska 2023], do these nuclei have more DNA and less DNA repair capacity?

Referee #3:

Referee report EMBOR-2024-59495-T
Immune cells dynamically adapt to confinement in vivo to enhance nuclear plasticity for migration

Summary:

Cell migration is essential for many physiological processes, and in particular in the case of the immune system, where immune cells need to navigate all types of tissues to perform immune surveillance and be recruited to sites of injury or infection. This process has been studied in vitro for many years, with researchers incrementally complexifying the in vitro reconstituted environments to mimic the constraints and signals cells might receive in vivo and understand how it affects their state and migratory abilities.

In this manuscript, the authors propose to use *Drosophila* as a model to study immune cell (leukocytes) migration in vivo, taking advantage of the numerous genetic tools, the ease of imaging in *Drosophila* and of the evolving environment faced by the leukocytes as the pupa wing is developing and vessels are forming. They propose to use this model to investigate the changes that occur in leukocytes as their environment is changing, for example when they become more confined, and how these changes could help leukocyte navigating the complex environments they encounter in vessels and upon extravasation towards an infection site. The authors state that by adapting the Lamin B/Lamin A ratio over these different stages (increasing Lamin B), leukocytes get an increasingly resilient nucleus that helps them navigating within tissues while avoiding nuclear envelope rupture or micronuclei formation.

The manuscript makes interesting observations on the evolution of the geometry of the environment available to leukocytes as the pupa wing develops, and of their nuclear lamina. They also highlight the dynamic deformations of their nuclei at the different developmental stages (18 h/ 40 h/ 75 h APF), reporting the formation of micronuclei. A major shortcoming is that the changes observed could be due to environmental changes other than confinement (chemical signaling, state of the surrounding epithelial cells, adhesion molecules...etc), or even to an intrinsic evolution of the leukocyte population, that could still follow some kind of developmental process. In this case, it is hard to be fully convinced that it is, as stated in the title, that leukocytes "adapt to

confinement", when it could be co-occurring evolution of the environment and the leukocytes, that could have been evolutionarily selected to optimize leukocyte migration at each step.

Another main claim of the manuscript is that this model opens the possibility of investigating cell migration and cell and nuclear deformation in different, complex environments *in vivo*. While this is true, and *Drosophila* offers many genetic tools and much easier experimentation than mouse models, one should not forget the work that can be done in mice (see Yan 2019, *iScience* for example for lymphocytes). In addition, other people have exploited *Drosophila* to investigate the importance of nuclear properties for cell migration in complex environments *in vivo* (Penfield, Montell, 2022, *J.Cell.Biol*, cited in the manuscript), which should be acknowledged.

The focus of the biological message of the manuscript is the study of the relative regulation of dLamB vs dLamC, and that the same population of leukocytes could regulate the expression of both lamina proteins over time to adapt to their environment. While this hypothesis is nice and could be correct, some experiments should be better controlled and/or interpreted.

In particular:

- Figure 1P: One pupa at 18h strongly differs from the others, which all have values that seem very close to the ones at 40h and 75h. Could you make this result stronger by including more pupae? Could this result be biased? Did the authors take into account the same number of single cell points for each pupa?
- In Figure S1J and Figure 1T, the authors use the photoconvertible fluorophore Kaede to assess whether the same leukocyte population is detected at different time points. However, could the converted fluorophore be transmitted to daughter cells if leukocytes proliferate? In which case, it would be the presence of daughter cells that is also detected, for which the state at the latest stages (75h) studied would not depend on their own experience of the previous stages (18h/ 40h for example).
- Figure 3, in the quantification of the *in vivo* images, the fluorescence intensities are normalized to the average epithelial intensity. However, it is shown in the supplementary figures that the epithelial cells do not have the same intensities at each time point, be it due to different treatment of the samples or to biological differences (which would highlight that not only the geometry, but many other factors change between these time points). Also, the background seems to be changing, maybe due to autofluorescence of the tissue, which should be measured. I do not believe the intensities can be compared between 18h, 40h, and 75h if one wishes to compare the amount of LaminB or LaminC. Since the *in vivo* and the *in vitro* results are also not strongly consistent, the authors could maybe turn to other techniques (WB, smFISH, short lived fluorescent protein under the reporter of dLamB or dLamA..).
- In the experiments comparing WT, dLamB-RNAi, dLamA OE and dLamA-RNAi: the legends mention using UAS-dLamB-RNAi, but this line is not mentioned in the methods (only dLamB-RNAi without UAS is mentioned). Is this a mistake? If the line for silencing dLamB is without UAS, there is a strong nuance to be brought to the interpretation of the data, as changing the Lamin B composition of the entire tissue will certainly affect the overall environment in which leukocytes are migrating. If this is not a mistake, but is due to the presence of the UAS-dLamC-RNAi, WT control should be Gal4? The effect of dLamC-RNAi is surprising as it goes in the same direction than the effect of dLamB-RNAi. This does not really fit with the idea that their ratio is important and that they compensate each other (unless what we see could be an effect of Gal4?). The data presented in Figure 4 regarding migration in these different genotypes are not very clear.

Minor remarks:

- Please add demarcations (white lines?) between videos in your supplementary movies
- Figure 1M and 1S: why measure the diameter at the widest point for the cell, but the minimum for the nucleus?
- Cell segmentation: given the image quality in movie S2, and the density of cells, I do not understand how you were able to segment and analyze cell shape robustly from this data. Did you select only isolated cells as the ones illustrated in Figure 1L?
- Figure 1H: Are the points averaged values over several fields of view, and each point is a pupae? Please precise this
- Figure 2K, Movie S5: There is a lot of background noise, I do not feel like the Myosin II-rich protrusions are very convincing
- Figure 2N-P: what are the blue vs green lines? they don't go in the same direction, and everytime only one (not always the same) goes in the same direction as the mean in black. This is unclear. How many cells were considered?
- Figure 2R-U, Figure S2O: It would be good to have more data points on the nuclear shape factor at 18h with no phagosomes, as this condition has far fewer data than the other ones, and the values are very spread. In general, to state the effect of phagosomes on nuclear shape, I believe that either phagosomal area or phagosome number (or both?) should be plotted against nuclear shape factor.
- In the experiments using the shibire construct, could the effect come intrinsically from temperature? What would happen if the authors were to put a WT pupa at 18{degree sign}C vs 32{degree sign}C?
- A table recapitulating the genotypes used would be most welcome and make the understanding easier (i.e one column "dLamC OE" then in the next column the corresponding fly line)

Overall, this manuscript proposes a nice system where researchers could look at single immune cell migration *in vivo*, *in live*, in a complex environment, and in an animal model where many genetic modifications are readily available. They also show that this system allows a high throughput, with many cells being imaged and analyzed here, and described the formation of micronuclei in a physiological, non-pathological context. The manuscript shows a large panel of the type of events that could be imaged in such a system, such as propulsion by Myosin II contraction, extravasation, cell deformation, and many cell trajectories, which demonstrate the functionality of the *drosophila* pupa as a tool to study immune cell migration in complex environment at the cellular level. However, the main claim made in the title of the manuscript is insufficiently supported by the data.

University of
BRISTOL
Faculty of Biomedical Sciences
University of Bristol
University Walk
Bristol
BS8 1TD UK

21st November 2024

RE: Manuscript EMBOR-2024-59495-T

In summary, we have strengthened our description of the changes in leukocyte nuclear dynamics and lamina as the wing environment matures and becomes increasingly confined; this includes new *in vivo* live imaging of endogenous Lamins and quantification of leukocyte nuclear dynamics when vessel formation is blocked. We have validated that the observed nuclear changes are not associated with increased leukocyte ploidy and provided additional evidence that the leukocyte lamina changes are wing-specific rather than a body-wide developmental change. We have now included additional data to bolster our suggestion that phagosomes can also indent the leukocyte nucleus. We have also carried out additional analyses to investigate how lamina perturbation affects leukocyte extravasation from vessels. Finally, we have made extensive improvements to the manuscript text to improve clarity, reduce repetition and be more cautious in some of our interpretations. We are grateful for the reviewers' suggestions that led us to perform these additional experiments and quantification, which we feel have greatly improved the paper.

The text below provides a detailed response to individual reviewer comments with our response to each suggestion/comment highlighted in bold following the text of their reviews. We have also highlighted each individual revision within the updated manuscript in yellow.

Reviewer #1:

The study by Karling and Weavers "Immune cells dynamically adapt to confinement *in vivo* to enhance nuclear plasticity for Migration" examines a very interesting and emerging topic in the field of cell biology and immunology. They address how changes in the nucleus affect immune cell migration in confined environments. Over recent years, this topic has mainly been studied in microfabricated devices, in particular microchannels with and without constrictions, where mouse or human immune cells moved in a confined space between two stiff surfaces. In these environments, the nucleus as the largest and stiffest organelle was shown to have rate-limiting roles for migration. Here, the authors present a novel, very elegant and much more physiological model system to address questions about the nuclear involvement in immune cell migration. The authors study hemocytes in the *Drosophila* wing system. As presented in this manuscript, this is a beautiful model system, allowing the observation of the same hemocytes over time at different stages of confinement, which result from the developmental stages of the wing vasculature. The authors provide wonderful and convincing imaging of hemocyte migration, including detailed imaging and analysis of nuclear morphology and integrity. Molecularly, the authors investigated expression changes of lamin B and A/C isoforms, showing that fine-tuning of the leukocyte nuclear lamina by balancing A- and B-type lamins aids nuclear deformation and efficient migration. Some interesting findings, e.g. phagosomes can deform nuclei even in non-confined environments or lamin B-deficiency results in decreased DNA damage (Fig. 5P), are presented but remain anecdotal. Instead, the authors show other potential use of this model system for studying nuclear squeezing during the process of extravasation (Fig. 6) or studying the influence of cell-cell deformations in confined environments (Fig. 7).

We thank the Reviewer for their overall positivity. As the Reviewer suggests, our intention in this manuscript has been to present the *Drosophila* pupal wing as a new *in vivo* model system

to study the diverse factors regulating immune cell nuclear dynamics. We envision that mechanistic studies on the role of intracellular phagosomes or links between nuclear Lamins and DNA damage will form the focus of in-depth future work.

Overall, I enjoyed reading this manuscript since it presented and discussed the findings of this study very well in the context of current concepts and perspectives of immune cell migration. Figures are executed very well, methods described very well, videos show exemplary cell dynamics of static data represented in the figures. Although not all observations were mechanistically followed up, the paper gives thought-provoking impulses, such as considering the influence of deformability in cells that shape physiological confinements, which cannot be performed very well in existing in vitro systems.

We thank the reviewer for their positivity. As the reviewer suggests, we use this study to present a new model system, highlighting key advantages and potential to enable more detailed mechanistic analyses in the long-term.

Major comment:

Since the paper is more descriptive than mechanistic, I would have wished that the authors also included some data how depletions of A- and B-type lamins influence the extravasation process, where nuclear squeezing and shape changes are particularly important. Since this has not been addressed in mammalian systems in detail, this study could provide important novel insight into this process from the *Drosophila* hemocyte perspective.

We have now provided additional data addressing the role of A and B type Lamins on extravasation efficiency (Figure 6L and Appendix Figure S3D; see text page 16).

Minor points:

1) Although the paper is really enjoyable to read, it is partly also a bit lengthy. The experimental results are presented in the light of previous research, explaining the rationale of the experiments, which is really nice. However, it feels as if some of these parts were redundant with parts in the introduction and discussion. A bit crisper version of the manuscript would be the presentation of a very good study even better.

We agree and have now modified the text accordingly, particularly in the results, to make the manuscript more concise.

2) In Figure 3I: The authors include "stiffer" and "softer" into the figure. Since this is probably based on assumptions from the literature rather than on real measurements, I would suggest to delete it from the figure.

As the Reviewer suggests, this labelling was based on assumptions from previous literature; whilst we speculate about these trends in the text (page 12), we have now removed the labelling from Figure 3 accordingly.

Reviewer #2

In vivo imaging of nuclear changes in shape and integrity have been a major challenge to study and relate to lamin levels that possibly change across a key developmental or disease timescale. The hemocytes and epithelial cells studies here provide such an opportunity, and the compelling and convincing studies here add in vivo relevance in the very basic fly model system to various past studies, mostly on isolated cells or tissue. The findings here will be a valuable addition to the literature after addressing a few concerns that temper enthusiasm.

We thank the reviewer for their overall positivity.

1. Fig.3's analyses of lamin levels indicate the B-type lamin (dLamB) is more variable with time but closer to epithelial levels (in solid tissue) than the A-type lamin (dLamC) that is much lower than

epithelial levels. These observations for hemocytes seem similar to those in Shin et al [PNAS 2013: Lamins regulate cell trafficking and lineage maturation of adult human hematopoietic cells] who reported that (i) solid tissue cells (e.g. MSCs) also tend to have much higher lamin-A levels and that (ii) 'B-type lamins vary by ~30-fold whereas lamin-A varies much less'. This relation to human blood lineage cells should be discussed and referenced by the present authors.

We apologise for the omission. Whilst we had referenced Shin et al in our original manuscript, we have now discussed its relevance in more detail in the text (page 12).

2. In human cells, LMNB1 increases 2-fold from G0/G1 to G2/M [Vashisth PNAS 2021], and so measurements of DNA content in the fly system at the later versus earlier timepoints need to be made and considered in relation to the 2-fold increase in dLamB. More DNA in the later nuclei could occur and also explain why there is more DNA damage measured per nucleus; in other words, more DNA provides more substrate for more damage and/or triggers a cell cycle checkpoint [e.g. Pfeiffer MBoC 2016]. Measurements of DNA per nucleus might be made from existing data for DNA or histone intensities.

To address this possibility, we have taken a two-pronged approach. Firstly, we have investigated the cell cycle state of the leukocytes at both 18h and 75h using the Fucci reporter (Figure EV3G-I and text page 11); we find that leukocytes at both developmental stages are at G0/G1. We have also assessed leukocyte DNA content through quantification of Histone H2A intensity (Figure EV3E-F and text page 11).

3. Fig.S5D shows more DNA damage after irradiation only at late times and only for LamB knockdown. Given that these nuclei tend to rupture more per Fig.5N and rupture releases DNA repair factors [Ivanovska 2023], do these nuclei have more DNA and less DNA repair capacity?

We apologise for the confusion here. To increase clarity, we have now displayed all data related to this point (originally in Figure 5 and Figure S5G-H) on a single graph (Appendix Figure S2I). Our data show that irradiation causes significantly increased DNA damage at both time-points (18h and 75h), and this occurs for both control and LamB knockdown leukocytes; in fact, irradiation-induced DNA damage does not differ significantly between control or LamB knockdowns (at either age). Without irradiation, LamB knockdown results in less DNA damage than controls (as detected using the γ H2AvD) but only in older 75h pupae.

It is interesting to consider that LamB knockdown leukocytes, which experience more nuclear ruptures, may possess less DNA repair capacity – we have now discussed this in the text (page 15) and included the [Ivanovska 2023] reference.

Reviewer #3

The manuscript makes interesting observations on the evolution of the geometry of the environment available to leukocytes as the pupa wing develops, and of their nuclear lamina. They also highlight the dynamic deformations of their nuclei at the different developmental stages (18 h/ 40 h/ 75 h APF), reporting the formation of micronuclei. A major shortcoming is that the changes observed could be due to environmental changes other than confinement (chemical signaling, state of the surrounding epithelial cells, adhesion molecules...etc), or even to an intrinsic evolution of the leukocyte population, that could still follow some kind of developmental process. In this case, it is hard to be fully convinced that it is, as stated in the title, that leukocytes "adapt to confinement", when it could be co-occurring evolution of the environment and the leukocytes, that could have been evolutionarily selected to optimize leukocyte migration at each step.

This is an interesting and important point. We have now discussed these concepts in the manuscript text (e.g. page 21), been more cautious in our interpretation of our data (placing more emphasis on the changing 'vessel microenvironment' as a whole rather than increasing

confinement alone) and have also changed our manuscript title accordingly. We have also provided additional data to address these hypotheses. Firstly, we have investigated whether the leukocyte lamina composition changes we observe is specific to the wing or rather reflects a general developmental change (i.e. occurs in leukocytes in other body tissues); we find that the lamina changes observed in wing leukocytes do not occur in other leukocytes elsewhere in the pupa (Figure EV3J-L and text page 11), suggesting lamina modulation is specific to the wing leukocytes. Secondly, we have compared leukocyte nuclear dynamics in wings of the same developmental age where vessel formation has been blocked (by epithelial-specific RNAi) (Figure EV2H-J and text page 8).

Another main claim of the manuscript is that this model opens the possibility of investigating cell migration and cell and nuclear deformation in different, complex environments *in vivo*. While this is true, and *Drosophila* offers many genetic tools and much easier experimentation than mouse models, one should not forget the work that can be done in mice (see Yan 2019, *iScience* for example for lymphocytes). In addition, other people have exploited *Drosophila* to investigate the importance of nuclear properties for cell migration in complex environments *in vivo* (Penfield, Montell, 2022, *J.Cell.Biol.*, cited in the manuscript), which should be acknowledged.

We have now expanded our discussion of these other *in vivo* models, ensuring all of these suggested references are now clearly cited in the text (page 4).

The focus of the biological message of the manuscript is the study of the relative regulation of dLamB vs dLamC, and that the same population of leukocytes could regulate the expression of both lamina proteins over time to adapt to their environment. While this hypothesis is nice and could be correct, some experiments should be better controlled and/or interpreted.

We thank the Reviewer for these suggestions and have responded to individual comments below.

In particular:

Figure 1P: One pupa at 18h strongly differs from the others, which all have values that seem very close to the ones at 40h and 75h. Could you make this result stronger by including more pupae? Could this result be biased? Did the authors take into account the same number of single cell points for each pupa?

We can see the pupa at 18h that the Reviewer refers to. We have now included additional 18h data in Figure 1P and ensured that equal cell numbers are provided for each 18h pupal dataset; nevertheless, these data updates do not alter the overall statistical significance of the results. Despite this, we do not think the environmental changes from 18h to 75h impact significantly on leukocyte speed and we have now stated this point more clearly in the text (page 7).

In Figure S1J and Figure 1T, the authors use the photoconvertible fluorophore Kaede to assess whether the same leukocyte population is detected at different time points. However, could the converted fluorophore be transmitted to daughter cells if leukocytes proliferate? In which case, it would be the presence of daughter cells that is also detected, for which the state at the latest stages (75h) studied would not depend on their own experience of the previous stages (18h/ 40h for example).

Whilst in theory the photoconverted fluorophore Kaede could be transmitted to daughter leukocytes, our analysis suggests that leukocytes rarely proliferate once vessels have formed (Figure EV1M). Nevertheless, we have now been more cautious in our interpretation of these data (text page 7).

Figure 3, in the quantification of the *in vivo* images, the fluorescence intensities are normalized to the average epithelial intensity. However, it is shown in the supplementary figures that the epithelial cells do not have the same intensities at each time point, be it due to different treatment of the samples or to

biological differences (which would highlight that not only the geometry, but many other factors change between these time points). Also, the background seems to be changing, maybe due to autofluorescence of the tissue, which should be measured. I do not believe the intensities can be compared between 18h, 40h, and 75h if one wishes to compare the amount of LaminB or LaminC. Since the in vivo and the in vitro results are also not strongly consistent, the authors could maybe turn to other techniques (WB, smFISH, short lived fluorescent protein under the reporter of dLamB or dLamA..).

The reason for performing both ex vivo and in vivo immunostaining of the dLamB and dLamC was to control for the issues that the Reviewer has identified.

We based our conclusions on how the absolute levels of leukocyte dLamB and dLamC change at different ages using our ex vivo data (Figure 3E-I) – where we can precisely control the experimental conditions and reliably compare the absolute levels of dLamB and dLamC protein between leukocytes isolated at different stages. This ex vivo data shows that absolute levels of dLamB are consistently higher in 40h and 75h leukocytes, compared to leukocytes from 18h (Figure 3H). The ex vivo analyses also suggests that levels of dLamC in leukocytes do not vary as much with age, although there is an initial increase followed by a more pronounced decrease (Figure 3G).

In contrast, the in vivo staining was performed to assess how the levels of leukocyte Lamins compared to that of the surrounding epithelial cells. As the Reviewer suggests, since the environment of the wing is changing, immunostainings worked to different efficiencies at different development stages. For this reason, we believe the in vivo data can only be used to compare the *relative* Lamin levels between leukocytes and epithelial cells (Figure 3L and 3M). We have not used this in vivo data to compare *absolute* levels of Lamins in leukocytes or epithelial cells across different stages (absolute levels were instead addressed using the ex vivo stains, Figure 3E-I). When performing the in vivo immunostains, we also measured the background fluorescence (Figure EV3D).

In summary, the ex vivo stains (Figure 3E-I) demonstrate the changes in *absolute* levels of Lamins within leukocytes across developmental stages. The in vivo stains, in contrast, demonstrate how the *relative* levels of lamins between leukocytes and epithelial cells change across stages (Figure 3J-M). We have now made this distinction clearer in the text (pages 11/12).

Nevertheless, to provide additional evidence to support our conclusions, we have provided new data from live-imaging of the endogenous dLamC-GFP and dLamB-RFP reporters, to complement the ex vivo and in vivo immunostaining data (Figure 3N-Q and Figure EV3M-N; discussed in the text pages 11/12).

In the experiments comparing WT, dLamB-RNAi, dLamA OE and dLamA-RNAi: the legends mention using UAS-dLamB-RNAi, but this line is not mentioned in the methods (only dLamB-RNAi without UAS is mentioned). Is this a mistake ? If the line for silencing dLamB is without UAS, there is a strong nuance to be brought to the interpretation of the data, as changing the Lamin B composition of the entire tissue will certainly affect the overall environment in which leukocytes are migrating. If this is not a mistake, but is due to the presence of the UAS-dLamC-RNAi, WT control should be Gal4 ? The effect of dLamC-RNAi is surprising as it goes in the same direction than the effect of dLamB-RNAi. This does not really fit with the idea that their ratio is important and that they compensate each other (unless what we see could be an effect of Gal4 ?). The data presented in Figure 4 regarding migration in these different genotypes are not very clear.

We apologise for the confusion here. All of the RNAi lines (for dLamB and dLamC) and over-expression lines (for dLamC) are UAS lines, which are driven specifically in leukocytes using

the *srp-Gal4* construct. All of the controls (labelled 'WT') for these experiments are thus the *Gal4* line alone without the accompanying *UAS-RNAi* or *UAS-over-expression* line. We have now made this clearer in the text (pages 12/13) and Figure legends.

We were also surprised that *dLamC-RNAi* produced an effect on hemocyte behaviour (e.g. nuclear dynamics) given that levels of *dLamC* appeared very low within the leukocytes (*dLamC* could not be detected in leukocytes by RT-qPCR or live-imaging of *dLamC-GFP*, it could only be reliably detected using *ex vivo* immunostains). We apologise for the confusion regarding our mention of 'compensation' (page 13); we did not intend to suggest the two different Lamin types compensate directly for each other (i.e. perform redundant roles). Rather, our analysis of Lamin levels following genetic perturbation suggest that leukocytes can dynamically adjust the expression of each Lamin type(s) when the level of another Lamin is perturbed. We envision that both Lamin types are required within leukocytes for optimal nuclear dynamics and cell migration. We have now modified the text to make this section clearer (page 13).

Minor remarks:

Please add demarcations (white lines ?) between videos in your supplementary movies
We have now added these lines.

Figure 1M and 1S: why measure the diameter at the widest point for the cell, but the minimum for the nucleus?

These measurements were chosen to address different questions. We envisioned that the 'widest/maximum cell diameter' would indicate the space occupied by the leukocyte cell body, to determine the proportion of the extracellular (luminal space) occupied (e.g. in relation to the overall vessel width). The 'minimum nuclear diameter' would indicate how small (and thus squeezed/elongated) the nuclei became. We have now performed additional analyses and included quantification of the maximum nuclear diameter (Figure EV1J, measured perpendicular to the vessel width) and minimum cellular diameter (Figure EV1I, measured across the main cell body excluding filopodia/lamellipodia).

Cell segmentation: given the image quality in movie S2, and the density of cells, I do not understand how you were able to segment and analyze cell shape robustly from this data. Did you select only isolated cells as the ones illustrated in Figure 1L ?

We apologise for the confusion here. We did not specifically quantify cells that appeared isolated within the z-stack projections, as we envisioned this would introduce bias into our analyses. Cell boundary detection was aided by referencing to the original z-stack (individual z-slices); we have now included this information in our Methods (text page 25).

Figure 1H: Are the points averaged values over several fields of view, and each point is a pupae? Please precise this.

Yes, as the Reviewer suggests, each point is a pupa; we have now made this clearer in the Figure legend text.

Figure 2K, Movie S5: There is a lot of background noise, I do not feel like the Myosin II-rich protrusions are very convincing

Unlike many of the other images in the manuscript, in which nuclear or cytoplasmic reporters are specifically expressed in the leukocytes, the images in Figure 2K are generated using endogenously labelled Myosin-II, the 'background' is thus Myosin-II present in surrounding cells (including the vein wall and epithelium). Nevertheless, we have now used z-projection images generated using fewer individual z-slices, which we think show the rear-localised Myosin II-rich

punctae more clearly; we agree it remains difficult to observe the Myosin II-rich cytoplasmic protrusions at the leading edge and have now adjusted our text accordingly (text pages 8/9).

Figure 2N-P: what are the blue vs green lines ? they don't go in the same direction, and everytime only one (not always the same) goes in the same direction as the mean in black. This is unclear. How many cells were considered?

We apologise for the confusion here, we have now made the description clearer in the Figure legend. Each line represents the linked data for a single cell. Lines have been colour-coded based on the gradient; lines with a negative gradient are depicted in blue and lines with a positive gradient are depicted in green. Cell (N) numbers are included in the Figure legend.

Figure 2R-U, Figure S2O: It would be good to have more data points on the nuclear shape factor at 18h with no phagosomes, as this condition has far fewer data than the other ones, and the values are very spread. In general, to state the effect of phagosomes on nuclear shape, I believe that either phagosomal area or phagosome number (or both?) should be plotted against nuclear shape factor.

We have now included additional data for nuclear shape factor, particularly for leukocytes lacking phagosomes (Figure 2X-Y). Since the majority of control leukocytes at 18h contain phagosomes, the 'no phagosome' leukocytes must be generated by expressing dominant negative Dynamin at 32°C (Shibire). As the Reviewer suggests, we have also replotted the data to show how nuclear shape factor varies according to phagosome number (Figure 2Y, discussed in the text page 9).

In the experiments using the shibire construct, could the effect come intrinsically from temperature ? What would happen if the authors were to put a WT pupa at 18{degree sign}C vs 32{degree sign}C ?

We do not believe that the differences in shape factor we observe depend upon temperature. The average shape factor for leukocytes containing phagosomes (either indenting or not indenting) does not differ between pupae raised at 18°C or shifted to 32°C (Figure 2X). To mitigate any potential effects of temperature, we specifically compared leukocytes with varying phagosome numbers that originated from pupae with identical genotypes (all carrying the shibireTS construct) that experienced identical temperature regimes (i.e. shifting from 18°C to 32°C) (Figure 2X). We now discuss this more clearly in the text (page 9).

A table recapitulating the genotypes used would be most welcome and make the understanding easier (i.e one column "dLamC OE" then in the next column the corresponding fly line)

In accordance with the journals preferred "Structured Methods" approach, we have now included a Reagents and Tools Table (listing key reagents, experimental models, software, and relevant equipment and including their sources and relevant identifiers).

Overall, this manuscript proposes a nice system where researchers could look at single immune cell migration in vivo, in live, in a complex environment, and in an animal model where many genetic modifications are readily available. They also show that this system allows a high throughput, with many cells being imaged and analyzed here, and described the formation of micronuclei in a physiological, non-pathological context. The manuscript shows a large panel of the type of events that could be imaged in such a system, such as propulsion by Myosin II contraction, extravasation, cell deformation, and many cell trajectories, which demonstrate the functionality of the drosophila pupa as a tool to study immune cell migration in complex environment at the cellular level. However, the main claim made in the title of the manuscript is insufficiently supported by the data.

We have now provided additional data (see individual responses above) to bolster our claims that leukocytes within the wing adapt to the changing wing environment (which includes increased confinement) by adjusting their lamina composition. Nevertheless, we have also

suggested an alternative manuscript title that refers to “confined environments” rather than adaptation to “confinement” per se.

Dear Dr. Weavers,

Thank you for the submission of your revised manuscript to our editorial offices. I have now received the reports from two of the three referees that I asked to re-evaluate the study, you will find below. As you will see, both referees now fully support the publication of the study in EMBO reports. Original referee #2 was completely unresponsive to my invitations to re-assess the manuscript. However, after going through your p-b-p-response and the revised manuscript, I consider his/her points as adequately addressed.

Before I can proceed with formal acceptance, I have these editorial requests I ask you to address in a final revised manuscript:

- Please make sure that all figure panels (main, EV and Appendix figures) are called out separately and sequentially. Presently, there is a callout to Figure S3M-N (line 356), but Appendix Figure S3 has no panels M-N. Please check.
- Please order the manuscript sections like this, using these names:
Title page - Abstract - Keywords - Introduction - Results - Discussion - Methods - Data availability section - Acknowledgements (including funding information) - Disclosure and Competing Interests Statement - References - Figure legends - Expanded View Figure legends
- Please remove the section 'Reagents and Tools Table - See additional text file' from the manuscript text file.
- Please do not show titles (the text in grey bars) in the figures (main, EV and Appendix figures). Please only show data and necessary labels. Titles and explanations/interpretations should only be part of the legends or the manuscript text.
- Please check that the number "n" for how many independent experiments were performed, their nature (biological versus technical replicates), the bars and error bars (e.g. SEM, SD) and the test used to calculate p-values is indicated in the respective figure legends (main, EV and Appendix figures). Please also check that all the p-values are explained in the legend, and that these fit to those shown in the figure. Please provide statistical testing where applicable. Please avoid the phrase 'independent experiment', but clearly state if these were biological or technical replicates. Please also indicate (e.g. with n.s.) if testing was performed, but the differences are not significant. In case n=2, please show the data as separate datapoints without error bars and statistics. See also:
<http://www.embopress.org/page/journal/14693178/authorguide#statisticalanalysis>

If n<5, please show single datapoints for diagrams. Moreover:

- Please indicate the statistical test used for data analysis in the legends of figures 1M, N, O, P, R, S; 5M, N, O; 6L; EV2 J.
- Please note that in figures 7G there is a mismatch between the annotated p values in the figure legend and the annotated p values in the figure file that should be corrected.
- Please note that the box plots need to be defined in terms of minima, maxima and percentile in the legends of figures 2M, 6G, H; 7H.
- Please note that information related to n is missing in the legends of figure 7H.
- Please note that the error bars are not defined in the legends of figures 1M, N, O, P, R, S; 2D, E, H, I, T, U, W; EV1 G, H, I, J, L, M; EV2 J, O, R, S."
- Please note that the solid arrow heads are not defined in the legend of figure 1H, EV3 F. This needs to be rectified.
- Please note that the white arrows are not defined in the legend of figure 1H. This needs to be rectified.
- Please note that the open arrows are not defined in the legend of figure 2k. This needs to be rectified.
- Please note that the white arrows and open arrows are not defined in the legend of figure 3J. This needs to be rectified.
- Please add to each legend (main, EV figures, Appendix figures where applicable) a 'Data Information' section explaining the statistics used or providing information regarding replicates and scales. See:
<https://www.embopress.org/page/journal/14693178/authorguide#figureformat>
- Please move the figure legends in the Appendix below the figures.
- The nomenclature for movies is 'Movie EVx'. Please use this for the source file names, the titles in the submission system, the ZIP folder names (see below) and the callouts in the manuscript text file. Please remove the legends for the movies from the manuscript text file. Instead, please provide each legend in a separate readme.txt file, uploaded and ZIPed together with the corresponding movie file, so that we have one ZIPed folder uploaded per movie.
- The panels shown in Fig. 2K and Fig. EV2M show a partial overlap (the myosin pattern?). Please clearly mention and explain this re-sue in the respective figure legends.
- Thank you for providing the requested source data. Please upload this as one folder per figure (with all files for one figure in one folder and ZIPed together) and one folder for all the source data for the EV figures and one folder for the Appendix figures.

Please do not provide one Excel file with numerical data for main, EV and Appendix figures but separate files for the numerical data for each figure.

In addition, I would need from you uploaded separately:

I look forward to seeing a new revised version of your manuscript as soon as possible.

Best,

Referee #1:

The authors addressed all my comments.

Referee #3:

I would like to thank the authors for the extensive work performed to both answer the remarks and questions in the point by point, and integrate the requests and comments of all reviewers in the manuscript. I am satisfied with the way they answered my comments.

I think the manuscript is now suitable for publication, and presents both a new in vivo model to study in vivo migration, and inspiring results looking into leukocytes adaptation of their lamin content to evolving environments during development.

All editorial and formatting issues were resolved by the authors.

Dr. Helen Weavers
University of Bristol
School of Biochemistry
University Walk
Bristol BS8 1TD
United Kingdom

Dear Dr. Weavers,

I am very pleased to accept your manuscript for publication in the next available issue of EMBO reports. Thank you for your contribution to our journal.

Yours sincerely,
